# Landslide risk management analysis on expansive residential areas. Case study of La Marina (Alicante, Spain)

Isidro Cantarino[1]; Miguel Angel Carrion[2]; Jose Sergio Palencia-Jimenez[3]; Víctor Martínez-Ibáñez[4]

[1,2,4] Department of Geological and Geotechnical Engineering, Universitat Politècnica de València, Camino de Vera, s/n, 46071, Valencia, Spain
[3] Department of Urban Planning, Universitat Politècnica de València, Camino de Vera, s/n, 46071, Valencia, Spain

*Correspondence to*: Víctor Martínez-Ibáñez (vicmarib@trr.upv.es)

**Abstract.** Urban expansion is a phenomenon that has been observed since the mid-20th century in more developed regions. One aspect of it is the urban development of holiday resorts with second homes that generally appeared following world political stabilisation. This residential expansion has often happened with scarce control, especially in its early stages, allowing areas to be occupied that are not so suitable in terms of the environment, culture and landscape, not to mention the very geological risks of flooding, earthquakes and landslides. Indeed, the risk of landslides for buildings occupying land in zones at such risk is not a matter solely attributable to the geomorphological characteristics of the land itself, nor is it simply a question of chance; it is also due to its management of such land, generally because of a lack of specific regulations. This study aims to lay down objective criteria to find how suitable a specific local entity's risk management is by looking at the evolution of its urban development procedures. It also aims to determine what causes the incidence of landslide risk (geomorphology, chance, land management, etc.), and finally to suggest control tools for the public bodies tasked with monitoring such matters.

## 1 Introduction

Landslide risk evaluation, management and mitigation are aspects that have been dealt with profusely in recent decades in the literature specialising in such matters. There is a multitude of studies on these matters, notably the summary put forward by Dai et al. (2002) with a critical review of landslide research and the strategies for reducing damages and losses, as well as the relevant publications by Lee and Jones (2004) and Glade et al. (2006) with a multidisciplinary perspective on landslide management. The recent review of quantitative methods for analysing landslide risk by Corominas et al. (2014) is also very noteworthy.

It is important to consider that the risk associated with landslides is changing as a consequence of environmental change and social developments. Climate change, the increased susceptibility of surface soil to instability, anthropogenic activities, growing (and uncontrolled) urban development and changes in land use with increased vulnerability for the population and infrastructure as a result, all contribute to the change—and in most cases the increase—in the risk of landslide (Gallina et al., 2016).

Urban expansion is a phenomenon associated with an increase in living standards and improvement in transport, communication and services outside the traditional population hubs. Among the many aspects of this phenomenon being studied, there is one that stands out as absolutely essential: the organisation and regulation of this urban growth. Indeed, in the classic work by De Terán (1982), the desired approach to urban planning is described as the need to establish order in developing it, in view of the damage and inconveniences caused by spontaneous urban development.

It is clear that urban expansion using unsuitable planning aggravates the incidence of geological risks. Specifically, landslides are one of the most dangerous natural disasters in terms of their frequency and the seriousness of the damage they do, leading

to loss of human life and social infrastructure in practically the whole world, which has been increasing in recent decades (Lee et al., 2017; Sandić et al., 2017; Cascini et al., 2005).

One of the main causes that explain the rise in geological risks in residential areas is expansive urban development processes, with a growing trend observed in these risks on a global scale, especially as regards landslides (Zhou and Zhao, 2013). These

expansionary activities have significantly increased the pressure on the land and consequently its effect on the population due to the occupation of land unsuitable for residential buildings (Fernández et al., 2016). This situation indicates improper management of the land, caused by a lack of suitable zoning of risks that hinders good planning for the use of the land (Cascini et al., 2005; and Cascini, 2008).

In other words, residential land usage may be exposed to greater natural risks precisely because such phenomena are not included in urban planning. When such planning is properly applied, it may help reduce exposure to the risk within urban areas. Indeed, it is considered to be a powerful tool in helping efficient, equitable adaptation between land occupation and natural risks (Hamma and Petrişor, 2018; Macintosh, 2013). There is a plethora of references that agree on the link between landslides and urban development. In some cases, there are rules on uses in said circumstances but they have not been taken

into account, thereby allowing for illegal and irregular occupation, as happened in the region of Campagna, Italy (Di Martire et al., 2012). In other cases, the course of rivers has been changed as a result of an increase in urban land, leading to negative effects on landslides as seen on the coast of Genoa (Faccini et al., 2015) and the city of Doboj in Bosnia and Herzegovina (Sandić et al., 2017).

The great demand for residential land to develop tourism in particular has caused similar situations. One example is the case cited by Katsigianni and Pavlos-Marinos (2017) on the Greek island of Santorini. A similar situation is seen in Mengshan, China (Peng and Wang, 2015), where engineering measures have been introduced *a posteriori* in a mountain tourist resort with a high risk of landslides. So, when drawing up and implementing urban planning, these types of factors must be taken into account amongst many others in order to suitably regulate the territory and prevent disorganised urban sprawl.


Faced with this situation, which has been widely recognised around the world, there is a need for risk governance to be duly included in urban planning (Renn and Klinke, 2013), improving the resilience of urban developments implemented and their possible growth (Zhai et al., 2015). It is also necessary to carry out suitable zoning of the risks to help reduce disasters (Wang et al., 2008). The great challenge is faced precisely in applying urban governance, attempting to define effective systems and

tools adapted to the new context of natural risks (Birkmann et al., 2014). Some experiences have shown the need to include the population's participation in tackling this problem, encouraging the adoption of solutions and management of them, as mentioned by Gough (2000) in New Zealand.

For all these reasons, it is surprising that the effectiveness and results of management of zones exposed to landslide risk have

received less attention. In the end, it is not only necessary to know how to quantify and locate the risks, as well as to put forward steps to avoid or mitigate them, but also to lay down procedures that can determine whether the management by technicians and politicians is effective and if the risk has truly been mitigated.

Thus, the main goal of this work is to determine whether the pace at which zones at risk are being occupied has a point of

inflection where it begins to steadily decrease. This point of inflection should be the result of a comprehensive application of specific regulations for the land that hinder or restrict residential construction in that type of area. It is along these lines that this paper suggests control tools for the public bodies tasked with monitoring such matters.

Another significant aim of this work should also be noted: This involves differentiating correct management of the terrain (specifically addressing its occupation by residential housing) from management that can clearly be improved. In particular, considering the risk of landslide for residential housing, the possibility of said risk becoming stabilised is studied over the time series. In this case, the management can be deemed adequate.

Nevertheless, if the risk increases over time, then it can be attributed to improper management, which should be corrected. The aims of this work also include analysing this situation, as well as determining, what causes an increase in landslide risk, for example by considering geomorphological dynamics, inadequate land management, even bad luck, etc.

To ascertain the importance of these control tools, a case on the Mediterranean coast has been studied in this work. Significant construction of new buildings has sprawled along said coastline, flouting planning regulations and thus proving the complete inefficiency of such regulations in containing this phenomenon (Malvárez and Pollard, 2004). That is why it is essential to enforce the government regulations developed, as well as to activate pertinent control mechanisms to ensure compliance.

## 2 General methodology

### 2.1 Objectives

Given the background described above, it is necessary to determine the extent to which residential areas are at risk of landslides, to understand the causes of these risks and to improve the planning for them. The basis for this should be a study of the behaviour of the risk taken upon building them and the factors determining it. To do so, it is necessary to begin with a map of the risk distribution and the annual residential construction data in a long time series. By knowing this risk and construction data, one can estimate its progress over time and whether a greater or lesser relative risk is being taken. Specifically, it is understood that this evolution in risk must not be exclusively a matter of the land's orographic characteristics, or even of chance, but it should also be greatly influenced by the pertinent territorial management.

The first task to be carried out is to gather residential building data as an annual summary for each local entity into which the zone of study is divided. Studying temporal series can then provide a lot of dynamic information about the evolution of a set of data. The series do not have to follow constant growth patterns since, as will be seen, they may undergo seasonal and other changes. This happens especially in the main data series to be analysed, which is the evolution of residential construction over time. Of course, it is also affected by the vagaries of big economic cycles, but such supra-annual seasonality is not going to be studied in depth.

The second data set must arise from the geolocalised map of risk distribution. Normally, this is based on a landslide susceptibility map (LSM) that has been deemed stable during the period analysed. Indeed, the risk map is calculated based on the temporal nature of construction and must be approximately in sync with this process. Moreover, the occurrence of a landslide is generally linked to trigger mechanisms that respond to events subject to a specific return period. The probability calculation also uses feedback from the appearance of these events, whose frequency is being modified as a result of climate change. However, according to Gariano and Guzzetti (2016), the effects of climate change on the type, extent, magnitude and direction of the changes in the slopes' stability conditions, and on the location, abundance and frequency of the landslides, are not completely clear. In the end, climate change is not going to be taken into account specifically in this work.

The main goal of this research is to seek risk modification patterns throughout a time series in local entities (hereinafter referred to as urban administrative divisions, UAD). Three main, non-exclusive hypotheses are proposed that enable the causes of the evolution in risk to be explained via a specific line of reasoning:


1.  Random reason, with no clear reason explaining the phenomenon;
2.  Geomorphological land characteristics: slope, lithology, land cover, etc.
3.  Management by local or regional public bodies responsible for land planning.

Simple observation of the annual evolution of risk in a specific zone is not by itself very conclusive in determining whether it is due to one of the causes described above. The trend has to be connected to the evolution of construction, verifying the temporal correlation between the two series within a local entity and among neighbouring local entities. This aspect will be analysed in the section dealing with the evolution of risk.

In keeping with the objectives described, it is necessary to have two fundamental types of georeferenced data: the data on residential construction evolving over time, and the data concerning the risk as a result of occupying the land, with the risk's distribution over time being variable depending on the pace of construction.

Logically, it is necessary to have data on the residential plots or parcels, specifically the data on the built-up area of each
residential parcel (as "gross floor area", hereinafter GFA), year of construction and geographic location. These types of data are beginning to be easily obtainable in some countries thanks to the development of public access digital cadastres (USA, Australia, France, Germany and others), which also appear to be near completion in many others. Currently, such data in Spain can be downloaded sequentially by municipalities via the Spanish Cadastral Agency (DGC).

### 2.2 Risk evaluation

The quantitative risk evaluation is to be carried out by applying the known general equation of risk (1), which includes the terms *Hazard* or probability, elements affected and their value (*Exposure*), and the seriousness of the damage (*Vulnerability*), based on the classic definitions from the Office of the United Nations Disaster Relief Organization (UNDRO, 1979).

$$\text{Risk} = \text{Hazard} \times \text{Exposure} \times \text{Vulnerability} \quad (1)$$


The value of risk is generally calculated in monetary units (€), though other types of unit may also be used (built-up $m^2$, casualties, etc.). In this work, the type of risk analysed is economic loss due to landslide damage to residential buildings.

HAZARD: This is the probability of occurrence of a potentially damaging natural phenomenon such as a landslide within a
specific period of time in a specific area. Calculation of this is normally based on a susceptibility map. Specifically, for each level of susceptibility the hazard must be calculated in units of probability, for which it is necessary to turn to inventory data of landslides. These two types of probability—temporal and spatial—are in keeping with equation (2):

$$\text{Hazard} = \text{Spatial probability} \times \text{Temporal probability} \quad (2)$$


EXPOSURE: People, property, systems, or other elements present in hazard zones that are thereby subject to potential losses. Therefore, exposure indicates the extent to which the elements at risk are actually located in the path of a particular landslide (Corominas *et al*., 2014).

VULNERABILITY: The degree of loss to a given element or set of elements within the area affected by the landslide hazard. It is expressed on a scale of 0 (no loss) to 1 (total loss). Vulnerability is probably the most difficult aspect to assess, due to the complexity and the wide-ranging variety of landslide processes (Glade, 2003). Following a technical/engineering approach, the seriousness of the damage done is a function of the magnitude or intensity of the landslide and the studied building's capacity for resistance.

## 2.3 Temporal evolution of risk

The essential purpose of this work is to define a reliable, simple method that will enable the risk's dynamics to be described. One strategy would be to recognise if the risk taken increases or decreases at the same pace as the construction of residential buildings. It would seem logical that this variation of risk should be estimated not as an absolute value, but in relation to the volume of construction at a given time, ascertaining whether there is a temporal correlation between these two variables or not.

An ideal situation pattern can be put forward of working with a long series of at least 40-50 years, since the beginning of the urban development boom in a specific zone. Three main sections can be found in this series with two different risk management types.

Firstly, let us consider a suitable management type. In the early years of this example situation pattern, there is disorganised construction occupying the most profitable spaces, but at the same time in not very suitable areas from the point of view of geological risk and the impact on the environment and the landscape. During the intermediate section, the occupation of zones at risk begins to change pace as urban development legislation begins to appear, along with land planning, environmental awareness, etc. The last section sees a very clear drop in the pace of the risk's growth, as the land regulation restrictions contemplated are directly applied. This theoretical behaviour is shown in Figure 1.

**Figure 1. Theorical evolution of risk accumulated over time for a one-year series pattern**

However, a varying panorama of unsuitable or improvable risk can also be found (Fig. 1). This type of growth in risk can arise when the pressure to build residential housing is so great that spaces become occupied that do not have the optimal conditions in terms of location and which until then had maintained their natural characteristics. Building on such spaces may entail taking greater risks because safer terrains have already been used up. Hence, the great increase in risk in Section 3 (Fig. 1, "Improvable management" line), should not be admissible in proper territorial management, and it is thus essential to provide tools to demonstrate such anomalies as shown in this work.

For dynamic analysis of the data shown in Figure 1, the two main annual data series must be used; one based on the evolution of the residential built-up area, and the other on the risk affecting part of that built-up area. The former is the Gross Floor Area (GFA, in m$^2$), calculated every year $y$ based on cadastral parcel data (CPi) by means of equation (3):

$$\text{GFA}(y) = \sum_{i=1}^{n} \text{GFA}_{(CP)i} \ (3)$$

Once the value of GFA(y) has been obtained, the simple moving average of order 3 for each year $y$, [MAvGFA(y)], is applied according to equation (4):

$$MAvGFA(y) = mean\ GFA(y - 1, y, y + 1) \qquad (4)$$

Applying the general equation of the risk (see equation 1) gives the risk value (RV) in €for each cadastral parcel CP affected, in accordance with the susceptibility map (equation (5)):


$$RV_{CP} = H_{CP} \times E_{CP} \times V_{CP} \quad (5)$$

Similarly, the simple moving average is calculated for the risk value MAvRV(y) via equations (6) and (7):

$$RV(y) = \sum_{i=1}^{n} RV_{(CP)i} \qquad (6)$$

$$MAvRV(y) = mean\ RV(y - 1, y, y + 1) \quad (7)$$

A relationship is sought between the two series to explain the trend towards a model of residential construction with increasing, stable or decreasing risk with relation to the built-up area. It is proposed that the relationship between risk and the built-up area should be used as an indicator of the evolution of risk and the construction associated with it.


Within a specific period of time, in the two moving average series a monotonically increasing interval can be selected that is limited by the years $[y_1, y_2]$ where $y_2 > y_1$. Two functions are defined for the risk values and for the built-up area: $f(y) = RV(y)$ ; $g(y) = GFA(y)$.

It has been confirmed that the way growth in risk with time directly relates to the pace of construction is determined by the behaviour of the quotient between functions $f(y)$ and $g(y)$. Thus, for example, proposing two growth ratios rRV and rGFA during the chosen period, which are approximately constant and where rRV > rGFA, it is easily shown that the quotient function is growing. In the opposite case, rRV < rGFA, the quotient function is falling.

The adimensional (relative) Risk Ratio (RR) between years $y_1$ and $y_2$ is defined in the following equation (8):

$$RR(y_2, y_1) = \frac{\frac{RV(y_2)}{RV(y_1)}}{\frac{GFA(y_2)}{GFA(y_1)}} = \frac{rRV}{rGFA} \qquad (8)$$

To sum up, it is concluded that $f(y)/g(y)$ is a function whose growth slope is defined by the Risk Ratio value (RR) for the
chosen interval $[y_1, y_2]$. The different options are summed up in Table 1.

**Table 1. Characteristics of the Risk Ratio RR.**

It is preferable to use the absolute values from the relationship between RV and GFA in order to be able to compare their
magnitudes between the different municipalities. In addition, working with functions of accumulated values $RV_{acc}$ and $GFA_{acc}$, it is ensured that the two base curves are monotonically increasing for the entire period being studied. It is easily demonstrated that the quotient function of the accumulated series $RV_{acc}/GFAa_{acc}$ also meets the characteristics determined for the RR value in Table 1.

These annual values can be transferred to a graph showing the resulting curve in order to analyse its ascending or descending trend, Figure 2.

**Fig. 2 Curve trend of different types of Risk Ratio**

Equation 9 shows the calculation of the accumulated RR values for each year

$$RR\ (y) = \frac{RVacc}{GFAacc} = \frac{\sum_{i=y_0}^{y} RVi}{\sum_{i=y_0}^{y} GFAi} \quad (9)$$

This equation is applied for the entire time series available, always starting from an original year $y_0$.

In these quotient functions, a simple deterministic trend is going to be assumed. Two specific indicators can be extracted from
these functions. The first of these would be to calculate the trend of the curve RR(y) simply by means of equation (10), which gives the slope of the straight line *m* that joins the two points of the curve RR(y) between moments *s* and *t* with periods of *n* years.

$$mRR(t,s) = \frac{[mean\ (RRt...RRt+n) - mean\ (RRs...RRs+n)]}{(t-s)},\ \forall\ t > s\ (10)$$

These reference points should be located in the temporal series at the moments prior to and after decisive changes in land management policy. It can also be used at the start of the series in order to learn the behaviour of risk in the early years of residential expansion.

Within the analysis of the temporal series of risk, it is worth noting that it may also be important to study the synchronisation
of their peaks to explain certain types of behaviour. Firstly, this may be done among the different geographically neighbouring local entities. For example, a specific type of municipal management would stand out if big differences are found with the neighbouring entity, especially if their geomorphological characteristics are very similar. To do so, three causes can be put forward to explain an external temporal correlation among neighbours, which fit with the hypotheses put forward in 2.1:

1.   With a total lack of synchronisation and without demonstrating behavioural patterns, the cause must occur randomly as a result of not very notable effects that cannot be analysed globally.
    2.   With synchronisation among neighbouring entities, the cause must be due to geomorphological characteristics of the terrain, since they are autocorrelated by geographic proximity.
    3.   With differing synchronisation in nearby areas but with certain patterns of behaviour in wider areas, the cause must
275        be sought in the different ways of managing the land.

Secondly, the internal synchronisation of construction peaks with the risk peaks for the same local entity should coincide in time under theoretical conditions. However, another two situations may also occur, which show that the construction and risk assumed are not necessarily governed by logic. Their possible reasons could be:

    1.   Risk peak brought forward: Buildings with a greater level of risk may be of greater commercial interest (e.g. due to dominant locations with the best views) and are thus built sooner.
    2.   Risk peak delayed: Suitable parcels begin to become scarce after a period of intense building activity, so that the last buildings are in a worse location and thus a greater risk is assumed.

A global view of the process is necessary, together with a complete study of the temporal series. It is usually preferable to summarise it in specific indicators that directly reflect the situation of the comprehensive temporal series for each of the urban administrative divisions (UAD, municipality equivalent) into which the study area is divided.

These indicators enable direct comparisons to be made, and analogies and differences to be seen more easily between different UADs. To do so, variables should be used that are not affected by the area of the UADs analysed. One solution is to calculate specific variables distributed homogeneously over the land's area.

    The most relevant factor is without a doubt the RR, derived from the quotient function RR(y), calculated as a summary of the

complete series in equation (11):

$$RR = \frac{\sum RV(\text{€})}{\sum GFA(m^2)} \times 1000 \quad (11)$$

where ΣGFA and ΣRV are total values for the complete period per UAD.

    The Risk Ratio defined in (11) allows us to know how much risk has been assumed throughout the period under study and to

be able to relate it to the other territorial units by specifying if it is greater or lesser than the average for the zone of work. It is thus possible to highlight the units that are assuming an excessive risk.

    A summary of other indicators that can be calculated for each UAD is shown in Table 2.

**Table 2. Global indicators per UAD**

### 3 Case study: La Marina

    The work by Cantarino *et al.* (2014) emphasises that Alicante was the province most affected by landslide risk value on residential buildings in the Valencia Community region (Spain), with more than one million euros in 2005 and 2009 each. This is chiefly due to the coastal zones in the northwest of the province (La Marina administrative division) with a high demand for

housing, which is an area susceptible to higher landslide risks.

    Thus, the area selected for this study is located in this area of Alicante (south-eastern Spain) bordering the Mediterranean Sea (Figure 3). The area includes 50 municipalities, covers 1,335 km$^2$ and has a population of 201,442 inhabitants according to the 2011 census (Spanish National Institute of Statistics, INE). This population has seen a notable increase since the 1990s (over

50%) basically due to tourist activity, though today it has fallen to 171,826 inhabitants in the last census (INE, 2018) as a result of the economic crisis. It is a populated mountainous environment rising from sea level to around 1,500 m. Its profile is shaped by its proximity to the sea, with a river system that deeply dissects the territory.

    **Fig. 3. La Marina area. Location of some municipalities mentioned in the text.**


    La Marina is located in the province of Alicante, which is the Valencia Community region's province with the highest landslide rate per unit of surface area (Hervás, 2017). Its extensive mountainous orography reaches the coastal strip itself, which is not free from risk. This situation is aggravated by being highly attractive for tourism and its residential occupation

This territory is typical of urban expansion around the Mediterranean basin, which is becoming increasingly intensive and no longer necessarily fostered or supported by the main coastal cities (EEA, 2006). It is an example of the so-called "rural sprawl" generated by second homes for the local population (in some cases first homes too), and of "residential tourism" for people from northern Europe, who spend long periods on the Mediterranean coast. Although initially there was a move towards

recovery and restoration of traditional rural constructions, strong demand has led to a proliferation of new-build housing units
(Pardo-García and Mérida-Rodríguez, 2018).

## 3.1 Data used

BASIC MAPPING

The official maps from the Spanish Geographic Institute (IGN) provided the borders and areas for the municipal territories to calculate the UADs. They also gave the 5×5m DEM (Digital Elevation Model) to calculate the mean slope of each
municipality.

LANDSLIDE DATABASE

The national Spanish database for landslides BD-MOVES from the Spanish Institute of Geology and Mining (IGME) was used, which follows the INSPIRE regulations (Infrastructure for Spatial Information in the European Community). BD-
MOVES, created in 2014, is made up of two blocks or sets of georeferenced spatial information: one referring to the description of the intrinsic, relatively invariable characteristics of landslides, and another referring to different activity events that led to said landslides, including morphometrics, triggering factors, damage and other data.

The other source of data is the landslide map to 1:50 000 scale in vector format drawn up by the Valencia Government's
Regional Department of Public Works in the project entitled "Lithology, exploitation of industrial rocks and landslide risk in the Valencia Community" (COPUT, 1998). This map uses geological and geotechnical data from the IGME, 1:50 000 scale topographical maps and aerial photographs available at that time.

CADASTRAL PARCELS

The information referring to cadastral plots or parcels was obtained from the cadastral mapping available from the DGC according to European INSPIRE guidelines. This cadastral information is provided by interoperable services (WMS and WFS) and can be downloaded in three datasets: Cadastral Parcels, Buildings and Addresses. For this study, the former has been chosen because it contains the main item defining the building. Within this item, we can find the data necessary for each parcel: built-up area (GFA), year of construction and type of usage. Only functional and residential parcels have been used for the
series 1960-2017.

CLASSES OF SUSCEPTIBILITY TO LANDSLIDES

To calculate the level of hazard, the starting point was the landslide susceptibility map (LSM) drawn up in a previous study (Cantarino *et al.*, 2019). Its characteristics are: pixels of 25 x 25 m as the unit of surface area and the spatial-multicriteria
method (SCME) to weight the factors for obtaining the susceptibility values. The three significant factors used were: slope gradient, lithology and land cover.

Specifically, the thresholds of susceptibility classes defined by Cantarino *et al.* were used. These thresholds were obtained by means of objective, meticulous classification based on a ROC analysis (Receiver Operating Characteristic), which uses the
intrinsic variability of the data and is one of the first applications of this type of map.. For this study, the spatial probability for each class has been determined by comparing these susceptible areas with the ones indicated in the inventory. This information, together with the temporal probability, has enabled the hazard and finally the risk to be calculated

Table 3 shows the susceptibility levels established via the susceptibility indices (LSI) that define them, together with the
number of pixels affected.

**Table 3. Land Susceptibility Index (LSI) values for the classes under consideration**

Figure 4 with some data used is attached, indicating the three highest levels of susceptibility, together with the location of landslides according to the Spanish Geological Survey (BD-MOVES) and the areas with instabilities according to the Valencia Regional Government (COPUT).

**Fig 4. La Marina area. Susceptibility, landslides location and areas with instabilities**

**3.2 Implementation of the method**

Figure 5 shows the flowchart indicating the method followed, which is explained in the above sections. It involves analysing the evolution over time of the residential parcel areas and landslide risks assumed in the urban expansion period in the La Marina area of Alicante province, from 1960 to 2017. Within this interval, a period of intense construction activity can be seen between 2000 and 2008, followed by a period of slowdown caused by the general economic crisis that occurred at the end of the decade of 2000 and which has not yet clearly ended.

**Fig. 5. Flowchart of the work procedure.**

Following the method described, firstly the cadastral parcels with their built-up gross floor area (GFA) were analysed, and then it was seen how the latter evolved over time together with the surface area affected by landslide risk. The final calculation used one-year periods to summarise the values dealt with individually for each parcel and as a moving average, MAv of order 3, according to equations (4) and (7).

The process followed for risk evaluation was based on locating the peak value of the three high susceptibility levels (between 3 and 5 class, see Table 3) for each cadastral parcel, affected by a buffer of 20 m around it. All cadastral parcels of an area of less than 10 m$^2$ were eliminated beforehand. The risk per parcel was then calculated, based on its maximum LSI value. As mentioned, possible changes in some of the factors involved in calculating the risk (such as those due to climate change) are not taken into account. The variation in real risk that may arise due to these changes is considered to be of little significance and does not therefore affect the final results.

By applying the equation to calculate the Risk Value (RV) shown in (5) for each cadastral parcel, it is possible to calculate the risk in monetary units (€) at the 2018 value. The year of construction is not considered, since in general the value is for the cost of reconstruction at the current value if affected by a landslide.

HAZARD

For the temporal probability in equation (2) (see Fell *et al*., 2008), one has to turn to databases such as BDMOVES from the IGME, which indicates the landslides and the date. For the spatial probability, work has been done with the COPUT's risk mapping for the zone under study.

In BD-MOVES, 13 landslides over the last 20 years are listed, though 5 of them are small slips. Summarising, it is possible to estimate 8 landslides for this period, with an annual probability (Pa) of 8/20. This annual probability should be adjusted downwards by an adjustment factor of *Faj*, but this value has been maintained since the inventory is not complete and the landslides that have not been included should be accounted for (Lee, 2009). Said probability was calculated in equation (12).

$$Pa = \left(\frac{\text{number of recorded events}}{\text{number of years in the record}}\right) \times \text{Faj} \quad (12)$$


To calculate the spatial probability Ps, landslides were selected that appear in COPUT's aforementioned map (1998), describing their limits and cross-referencing this information with susceptibility levels 3, 4 and 5 of the map listed in Table 3. The results are shown in Table 4.

**Table 4. Probability of occurrence and associated hazard by susceptibility level**

Classes 1 and 2 of the susceptibility map (LSM) are not taken into account because they do not show a probability of being affected by risk of landslide. Thus, for each level L of the LSM shown in Table 4, the value of hazard level is obtained using equation (13).


$$H_{\text{CP}} = f(\text{LSImax}) = \frac{S_{\text{RL}}}{S_L} \times Pa = Ps \times Pa \quad (13)$$

Where $S_L$ is the total surface area of level L, and $S_{\text{RL}}$ is the surface area of level L affected by risk of landslides.

EXPOSURE

Only residential housing, which is generally terraced or detached, is to be considered as affected elements. Buildings or high-rise residential blocks are not built in the areas dealt with (with the notable exception of the municipality of Benidorm), but in areas that are generally flat and/or near the coast. The mean number of floors confirms this matter (see NFm in Table 2).

The value of these elements only takes into account the gross floor (built-up) area, not the value of land that is not affected by landslides. Taking into account only the cost of constructing the building to calculate the Building Execution Unit Cost, the tables of the Institut Valencià de l'Edificiació are used, IVE (website: http://www.five.es/, see online in *Productos/Herramientas*). To do so, the definition of Basic Building Module (BBM; €/ built-up m²) is used, which represents the material cost of implementation per built-up square metre of the Reference Building, implemented under conventional worksite conditions and circumstances.

The BBM for December 2018 for single-family detached houses of fewer than 3 floors with an inhabitable surface area of over 70 m² and with high quality finishings and fittings, is €829/m². This value remained practically constant throughout 2018, and even as of 2008 it has been above €800/m². Open-plan buildings of 3 floors or more, up to 80 homes and an inhabitable surface area of between 45 and 70m², are valued at €780/m². To a large extent, the homes affected are of the single family type, so the value of reconstruction has been taken to be constant at €800/m².


The value for reconstructing each cadastral parcel is calculated according to equation (14), without taking into account the value of the land.

$$E_{\text{CP}} (\text{€}) = \text{GFA} (m^2) \times \text{BBM} (\text{€}/m^2) \quad (14)$$


VULNERABILITY

In order to determine landslide magnitude (LM) in a geographical area, it is crucial to create a landslide inventory to know the main landslide types, landslide morphometric parameters, landslide velocity and observed damage. This data is not provided by the available landslide databases such as BDMOVES and COPUT.

In the La Marina area, the predominant failure mechanism for shallow slides is along the existing dip planes of the Cretaceous limestone geological formations. According to Fell (1994), these landslides are defined as small landslides. The shallow slides occurring in the study area are rapid landslides, according to the velocity scale proposed by Cruden and Varnes (1996), with a typical velocity ranging from 1.8 m/h to 3 m/min. In La Marina, damage or loss caused by past landslides is poorly documented and this is a major constraint in drawing up vulnerability curves. However, field observations have shown that shallow slides that have occurred in the study area did not have enough energy to completely destroy a building. Typical damage produced by shallow slides in the study area is shown by cracks opening up in the buildings' walls. This type of damage caused by landslides in buildings is classified by Leone (1996) at level III (from I to V), which corresponds to a structural damage of 0.4–0.6 on a scale ranging from 0 to 1. Taking into account the previous example and the fact that shallow slide characteristics in the study zone do not vary too much in terms of affected area, depth of the slip surface, velocity, volume and typical damage, we assumed a single fixed value for LM (in accordance with the level of susceptibility). Therefore, the LM was assumed to be 0.6 for the area of study on a heuristic scale ranging from 0 to 1 (Silva and Pereira, 2014) (see Table 5).

The other factor to evaluate the final vulnerability, FV, is to estimate the considered residential buildings' resistance (BR) taking into account the type, materials, age and height of the building (Kappes *et al.*, 2012). Within the zone under study, the construction techniques, materials used (mainly concrete) and structure are quite similar and are considered to be sufficiently resistant with a generally good state of conservation seen in the buildings. The biggest difference one can find is in the mean number of floors for each building, though the type of home affected has a low number on average (see NFm in Table 2) with not very significant variations.

Papathoma-Köhle *et al.* (2017) identify a list of indicators for one particular kind of landslide (debris flow) physical vulnerability assessment of buildings. One of them, the height of the building, directly influences the degree of loss. In accordance with Papathoma-Köhle *et al.*, the higher the building, the fewer the expected losses, so a greater BR is considered in these cases (Table 5).

Equation (15) enables the final vulnerability FV to be calculated, in which the BR depends solely on the number of floors NF, and Table 5 gives the values obtained by applying it.

$$FV = LM \times (1 - BR_{NF}) \quad (15)$$

**Table 5. Vulnerability related to the number of floors.**

RISK

The final calculation of risk for each cadastral parcel is the result of applying equation (3). Thus, in accordance with the equations shown above, the final expression for the calculation of the risk value for each CP is:

$$RV_{CP}(\mathbf{\in}) = \left( \frac{S_{RL}}{S_L} \times Pa \right) \times \left( GFA \times BBM \right) \times \left[ LM \times \left( 1 - BR_{NF} \right) \right] (16)$$

As a final reflection on the application of this or any other method for calculating risk, it should be noted that there is some difficulty in obtaining precise results due to the lack of official data and specific, up-to-date studies in the sphere being studied.

Some of these procedures are based on data that is not very exact, and even on subjective evaluations, which means some error must be assumed in the results obtained, though this does not invalidate the objectives or the validity of the index originally proposed in our study. For this reason, the global calculation of equation (16) has been carried out using the main factors without including ones considered to be less relevant.

**3.3 Risk curve and trend**

The year the temporal series begin is determined in Spain and in the Valencia Community region as 1960, which marked the start of tourism expansion on the Mediterranean coast. The year approximately coincides with when the Law on Centres and Zones of National Interest to Tourism was passed (year 1963), which notably fostered residential construction in coastal areas without taking into account geological risks.


As has been mentioned, to study the evolution of risk, the proposal is to use a complete analysis of the temporal series of the Risk Ratio value (RR) as the basis. Indeed, the shape of the RR(y) curve, as well as the behaviour of the two annual series of GFA(y) and RV(y), enable the characteristics of the evolution of risk to be established for the entire period.

When the RR(y) has been calculated, its three singular points are extracted to define the straight lines and calculate their slope via equation (10). Specifically, the mean points of the curve were used for the two different periods that include the decades 1960-69, 1980-1989 and 2000-2009 for a time interval of 20 years. The slopes calculated have been called *mRR Lo* for the lower (earlier) period (60s and 80s) and *mRR Hi* for the higher (later) period (80s and 2000s).

The first period analysed explains the historical evolution, marking the beginning of the trend, which is why the mean points have been selected from the 1960s and 1980s. For the second period, the decades of the 1980s and 2000s were used. This period acts as a reference for the substantial change in land policy, which should have brought about a clear change in trend. Indeed, it was in the 1990s that the first official study on the risk of landslides appeared (COPUT, 1998). Such work continued with legislative activity that fostered the prevention of natural or induced risk.

**4 Results**

The values of these indicators calculated for the 50 municipalities that make up La Marina are shown in Table 6, accompanied by their interval of variation. A series of annual values were calculated for the 50 municipalities of La Marina area as a whole. The total values for the built-up area (GFA) and risk (RV) are shown in Table 6. The mean values are listed in the same table, as well as their interval of variation of the global indicators in the previous Table 2


**Table 6. Total values and global indicators per municipality. For indicators, means and variation intervals.**

The values of these indicators can be explained logically and are subsequently used to classify the municipalities via a cluster analysis. On drawing up the graphs, the ratios between the total values of GFA and RV from Table 6 were used, which is 530 approximately 8:1 (GFA:RV).

As a result of the analysis of the RR, GFA and RV graphs (see the available research data), some interesting behaviour can be found. The comparative graphs of GFA and RV are particularly useful. In general, a marked stability can be seen in the final stretch of the last 10 years, possibly caused by the slowdown in construction after the 2008 crisis. This enables us to affirm 535 that acquisition of residential land with low risk has not been exhausted.

The annual risk peak values are also seen to appear usually after the construction peaks, or at least they are seen very clearly in municipalities with the greatest construction activity. Recalling the possible causes for this situation (listed in section 2.3), this may be due to the fact that after an intense construction period the last parcels to be allotted are usually in zones of greater

risk, since those of lesser risk have been allotted first. However, in municipalities with less construction, the construction peaks are more synchronised and even appear before the risk peaks.

Lastly, there is no synchronisation found between the different curves in neighbouring municipalities . Nevertheless, a few behavioural patterns have been obtained in the geographic area under study. Hence, as explained in 2.3 for the so-called internal

synchronisation, the most probable cause should be sought in the differing land management, and not in geomorphological or random causes.

Figure 6 shows the evolution of two neighbouring coastal municipalities that represent those with greatest residential construction with a slope close to the average, but which have very different characteristics in assuming risk. They are Calpe

and Altea (see locations in Fig. 3 and 4); the former with RR = 79.8 and the latter with RR = 463.3.

**Figure 6. Evolution of the annual series of GFA, RV and RR in the municipalities of Calpe (a, b, c) and Altea (d, e, f).**

Calpe is a mountainous coastal municipality with a high construction rate but a clearly low risk, with a lower risk than the

average according to Figure 6a. In terms of cumulative value, 6(b) also shows the construction as being more significant than risk, with a sharper slope for the former. Graph 6.c shows there is an early stage in the 1970s with a risk peak, which then gradually falls. The RR indicator is very low and everything seems to indicate suitable management over the last 20 years, taking on a comparatively low risk. This pattern is similar to "Suitable management" line of Fig. 1.

In Altea, on the other hand, a greater risk is seen to be assumed in the second half of the series, which is above average (6(d)). Moreover, Figure 6(e) shows risk more significantly than construction.  Figure 6(f) indicates an appropriate beginning for the RR value, but later the relative risk grows. As the indicator value is very high, it can be concluded that this municipality's management should clearly be revised, with a change in trend sought. In both cases we can see risk peaks that come after their corresponding construction peaks. This pattern is similar to "Improvable management" line of Fig. 1.


The possible explanation could be that the plots at greatest risk of landslide begin to be used at a greater pace once the best plots have been occupied following a period of intensive building activity. In other words, it is possible that when suitable plots become scarce, the next buildings are constructed in a worse location and thus a greater risk is taken on.

For the other municipalities, a similar criterion has been followed. High RR values and a straight line with an increasing trend in the second half of the period point to a necessary revision of the protocols in granting construction licences, in view of the growing risk assumed. On the other hand, RR values lower than the average coupled with a decreasing trend indicate a lowering risk and improved land management.

To conclude, Figure 7 shows the joint evolution of the whole La Marina area (excluding Altea and Benitachell due to the bias they would introduce). Figure 7(a) shows continual growth in construction and risk almost simultaneously, indicating a clear similarity with the curve pattern shown in Fig. 1 in the three intervals. These curves show a marked jump in the decade of 2000, coinciding with a period of clear economic boom associated with intense construction activity (known as the "Spanish

property bubble" from 1998 to 2008). Finally, 7(b) shows fast growth in risk during the first part of the period under

consideration, levelling out and becoming comparable to the growth in residential area in the second part of this period. To sum up, no generalised drop is seen in the risk growth rate, so it is hoped that in coming years the urban development regulations in force will end up serving their purpose.

**Figure 7. Evolution of the annual series of GFA, RV and RR for the La Marina area.**

**5 Discussion**

The analysis of the graphs for municipalities is very revealing in learning the effectiveness of their management in lowering the risk of landslides. However, it is important to observe how the municipalities studied are organised and what type of association there may be among them. To do so, a cluster analysis was applied in order to determine the types of groupings that can be found in the area being studied. This type of analysis is a tool that has widely shown its usefulness in grouping

urban areas by means of indicators (Huang *et al.*, 2007; Stewart and Janssen, 2014; Goerlich *et al.*, 2017).

The variables to be included in the cluster analysis as explanatory variables were the indicators for each municipality in keeping with Table 6. They are variables for the period from 1960 until today. However, some of them were discarded *a priori*. Firstly, this includes the mean number of floors NFm, as there is little variability in this. The mean distance to the historical centre,

Dhc, is intended to be a measurement of residential expansion, but it is excessively related to the size of municipality and the location of its historical centre, so that it was also used little. Lastly, mRR Lo was not considered as it is not a main variable and behaves as secondary in the current evolution of risk.

Hence, the variables initially selected for the cluster analysis were: Mean slope SLm, mRR Hi, SpGFA, Dc, SpRV, RRt, RRm,

previously standardised. Nevertheless, on carrying out an analysis of prior correlations to avoid variables that do not explain variance as a whole so much, it was found that SpGFA has a very strong linear relationship with the variables SLm, Dc, RRm and SpRV. This means that the rate of construction increases in flat and coastal areas, leading to less risk. Thus, it was decided to eliminate this group of variables from the cluster analysis.

Finally, the analysis was carried out only with the following indicators: the rate of built-up area SpGFA (in $m^2$/k $m^2$ of UAD), the total Risk Ratio RRt (€1000 $m^2$ GFA) and the final section of slope of the straight trend line mRR Hi (degrees). These indicators have proven to be sufficiently explanatory variables to be able to establish groups with homogeneous characteristics. In this analysis, all hierarchical methods were tested with different numbers of clusters. Ward's method and the Manhattan distance gave the best results. Various attempts were made to find the optimum cluster number of 10 and seeking to

differentiate two municipalities with clearly different behaviour: Calpe and Benidorm. Finally a solution was chosen with the greatest number of clusters in order to isolate these unique cases, with 14 in total.

The centroids of the 14 clusters have been organised into four sets A, B, C and D (from smallest to biggest in magnitude) according to the percentiles values (90, 60, 30 and 0%) of the three variables chosen for the analysis (see Table 7). These limits

thus established are particularly intended to restrict the upper values of the series (percentil >60% in A and B). It is thus possible to more clearly highlight the cases that should be addressed in order to manage risks properly. Four types of evaluation for risk building management have been defined for the final curve section value (mRR Hi).

Each of these classes is defined as the result of a new grouping into four clusters for each variable. Table 8 shows the cluster results organized by levels and includes two indicators that provide information relevant to the established clusters. Those two indicators are the mean slope (SLm in degrees) and the specific risk rate (SpRV in €/km2), previously defined in Table 2. Table 9 explains each cluster's most relevant characteristics and the municipalities within each of them, if the trend is different for the first section, then the name of the municipality is marked at the end with an asterisk (*).


**Table 7. Centroids classification values**

**Table 8. Clusters centroids and their levels. Organized from A (max) to D (min) according to Table 7**

**Table 9. List of clusters with their characteristics and municipalities assigned. Grouped by intensity construction ratio (SpGFA) from high to low**

As can be seen in Tables 8 and 9, the municipalities with High and Very High RR have been differentiated in italics. This situation only occurs with municipalities with a high rate of construction in zones at risk, or in inland too, with very
mountainous municipalities where residential buildings are positioned more easily in risk zones, leading to a higher RR. In municipalities with improvable management (BBA, DAA and DBB cluster codes), these high RR values seem to be due to the fact they have been taking on higher risk rates over the last 20 years than in the rest of the historical series. On the other hand, if there is suitable management (only DAD cluster code), these are municipalities that took on greater risks in their first 30 years than they currently do.


Table 9 is shown in map form in Figure 8. These maps show the municipal distribution of the groups obtained by means of the cluster analysis, as well as their specific risk values (SpRV).

**Fig. 8. Map of La Marina: (a) with cluster groups\*, (b) with the SpRV value. (\*) Clusters D1: DAA, DBB, DDB; Clusters D2: DAD,**
**DBD, DCD**

Based on the results obtained, it is seen that many of the municipalities with suitable management today began with overexposure of residential construction in risk zones (marked with "\*") in the early decades of the series. This is also seen in the Fig. 1 (Suitable management line) or in the fact that the mRR Lo value exceeds the mRR Hi value. This is logical because during the initial period the protection policies were not so developed. Taking advantage of this lack of control, together with
the urban development initiative proposed by tourism legislation, it was possible to construct a greater number of buildings in unsuitable areas.

In the maps in Fig. 8, the municipalities of cluster BBA (Altea and Benitachell) clearly stand out (see location in Fig. 4), where a growing occupation of risk zones can be seen. Indeed, according to Table 8, this cluster has the highest RR value of all the
municipalities in groups Axx, Bxx and Cxx, as well as its SpRV and the clearest upward trend. These are therefore the clearest examples of coastal municipalities that should review their criteria for considering land as apt for urban development. Both municipalities are in the process of reviewing their municipal approaches, since in their previous plans established in the aforementioned first period there were no limits established as regards use based on geological risks.

At the other extreme, there are the mountainous municipalities in groups Dxx (see the maps in Fig. 8) with a high mean slope (>17°). Their location can be clearly seen in the inland strip with broad unstable areas where it is more probable for construction to occur in them. This seems to be the reason why this group has many municipalities with a high RR value, sometimes

burdened with this since the beginning of the series due to the effect of homes in the urban hub itself. Furthermore, since these are municipalities with low populations (under 5,000 inhabitants), they do not usually have the means to draw up land regulation plans or the technical staff to update them.

Perhaps cluster DDA (Confrides, see location in Fig. 4) is the paradigm among these mountainous municipalities, since it has the second lowest construction rate but a high RR and a growing trend towards risk. It is also the one with the parcels furthest from the coast. It does not have land planning and according to COPUT (1998) it is one of the municipalities with the greatest density of landslides per unit of area in La Marina. Although this does not affect a significant number of homes in absolute values, in these conditions an improvement in the trend values and risk indicators would seem to be far off. Indeed, this is a singular case in which the municipality has not expressed urban development intentions in any of the three previous periods, but will necessarily have to adapt to the land planning regulations in force.

**6 Conclusions**

As a final reflection, it would seem reasonable to think that studies on the mechanics and distribution of landslides, the growth in information about behaviour of the ground, the restrictions imposed on residential expansion, etc., should progressively improve the effectiveness in tackling the risks. However, it has been shown that not all municipalities are capable of reducing the incidence of these risks over time and that, according to Fig. 8, this incidence is still generally high. So why is this happening?

In Section 2.1, three possible hypotheses have been put forward to explain this situation. Firstly, the analysis cluster does not enable a direct relationship to be seen between the land's geomorphological characteristics (mainly the mean slope SLm) and the variation in risk. In other words, there are contradictory cases. The same could be said about municipalities with a greater or lesser volume of construction, proximity to the coast, etc. Hence, a greater or lesser risk value and a growing or falling trend cannot be attributed to the intrinsic qualities of the municipalities studied. Nor can they be attributed to strictly random factors, since there is coherent behaviour within the clusters analysed.

The above conclusions are bolstered when one considers the lack of temporal correlation found for the data in neighbouring series, together with the existence of global behaviour patterns (see Section 2.3). As a result of all of this, the assumption of greater or lesser risk and its temporal evolution seems to be exclusively due to the third hypothesis initially put forward in the aforementioned Section 2.1; i.e. land management. In this study, procedures have been proposed that are based on analyses of graphs and risk indicators in order to find trends and behaviours that may subsequently help to improve this land management.

The Risk Ratio (RR) developed in this article stands out as a robust indicator for directly finding the relationship between residential construction and its associated risk. It is especially useful for coastal municipalities with a high rate of construction, since it differentiates between those that take on a higher risk than those that do not. Nevertheless, in municipalities located in the inland mountainous strip, with a low residential construction density, high susceptibility and which do not usually have land planning, the values are also high. In these cases it is not possible to strictly attribute these values to unsuitable management.

In general, it is seen that coastal municipalities are more prone to assume greater specific risk (see Fig. 8), although the pace of growth in risk is lower than for construction. In mountainous municipalities in the inland strip, precisely the opposite happens. Of course there are a fair number of exceptions to this rule, but two coastal municipalities especially stand out, where

their great construction intensity is exceeded by the growing pace of occupation of zones at risk. This is group BBA, which includes Altea and Benitachell (the characteristics of Altea have already been specified in Fig 6). Although their land occupation has not reached the level of Very High intensity construction ratio (SpGFA), both municipalities are characteristic for having High SpGFA and RR values, which shows a growing occupation of locations at risk.

Benidorm (ADC cluster) is precisely an example worth highlighting (Fig. 4). It is a coastal municipality that is internationally known as a holiday destination with a notably mountainous profile. It has one of the biggest construction rates in the area, but this has not led to occupation of extensive risk areas, although there is a slight upward trend. It is not surprising, then, that this is the only example of "vertical" construction, where the mean number floors per building (5.85) is significantly greater than in the other municipalities in the study (2.09). Hence, it can be considered a suitable policy if the objective is to provide a greater amount of built-up area in relation to the risk taken (RR = 86, found in 30% percentile).

To sum up, none of the basic risk parameters in any municipality seems to be determined by randomness, and only in the most mountainous ones is it determined by the orographic conditions of the land. Monitoring and restriction of building in risk zones must be applied mainly in the coastal municipalities with a greater rate of construction. Residential construction's avoidance of zones at risk of landslide will depend on the municipal technicians having complete, up-to-date information in their urban development regulation planning; in other words, they should have been reviewed in the last decade. Only in this way will it be possible to have objective criteria in order to enforce urban development regulations and their implicit "precautionary principle" in order to guarantee the greatest possible level of protection.

The risks of landslide are a result of human activity itself, and it is also of great human concern to minimise them. The mechanisms for monitoring and control that should be working to reduce them must not be solely the responsibility of the municipality, but also of public bodies of greater hierarchy that may ensure they are applied by using their best resources and regulatory capacity. Tools have been developed in this work to take objective decisions to suitably adapt land management, and this can be extended to other residential areas. Applying them does not guarantee that the problem will be eliminated, but at least it will help alleviate them and act as a guide to solve them.

*Data availability*. Borders and areas for the municipal territories and 5×5m DEM (Digital Elevation Model) are available on the Spanish Geographic Institute (IGN) website (https://www.ign.es/web/ign/portal). Database for landslides was processed from BD-MOVES, available on the Spanish Geomining Institute (IGME) (http://mapas.igme.es/gis/rest/services/BasesDatos/IGME_BDMoves_ES/MapServer), also from project entitled "Lithology, exploitation of industrial rocks and landslide risk in the Valencia Community", available online on http://www.cma.gva.es/areas/urbanismo_ordenacion/infadm/publicaciones/pdf/litologia/. The information referring to cadastral plots or parcels was obtained from the cadastral mapping available from the Spanish Cadastral Directorate (DGC) (http://ovc.catastro.meh.es/INSPIRE/wfsCP.aspx?)**.** Landslide susceptibility map (LSM) with a resolution of 25 x 25 m in La Marina, can be found in Cantarino *et al*.( 2019). Further information can be made available upon request to the corresponding author.

*Author contribution*. All authors contributed to conceptualization, leaded by IC, who also conducted formal analysis and initial draft. JS had a leading role on urban planning perspective. MA contributed to validation and data visualization. VM critically reviewed the paper and conducted pre- and post-publication stage.

*Competing interests*. The authors declare that they have no conflict of interest.

*Acknowledgements.* Authors acknowledge funding from Department of Geological and Geotechnical Engineering, Universitat Politècnica de València.

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

**Table 1. Characteristics of the Risk Ratio RR.**

| RR value | Type of curve | Characteristic | Discussion |
|---|---|---|---|
| >1 | Monotonically increasing | Growth rate of the risk value greater than that of the built-up area. | Disproportionate risk growth, without restrictions or planning. Unfavourable situation. This must occur in section 1 of Fig. 1. |
| ≈ 1 | Monotonically constant | Growth rate of the risk value similar to that of the built-up area . | No well-defined evolution. Situation not clearly favourable. |
| <1 | Monotonically decreasing | Growth rate of the risk value less than that of the built-up area | Growth of risk restricted, which must be due to some type of criterion. Favourable situation. This must occur in section 3 of Fig. 1. |

**Table 2. Global indicators per UAD**

| Name | Formula | Notes |
|---|---|---|
| Total Risk Ratio, $RRt$ (€1000 m$^2$ of GFA) | $RRt = \Sigma RV / \Sigma\ GFA\ \times 1000$ | Already commented in the text (Eq xx). |
| Mean RR, $RRm$ (€1000 m$^2$ of GFA) | $RRm = \Sigma\ RR/ny$ | Less useful as it is an average of averages. |
| Mean no. of floors per CP, $NFm$ | $NFm = (\Sigma\ (GFA/S_{CP}))/ny$ | Interesting to know the type of buildings in the UAD. |
| Mean no. of floors in CP affected by risk, $NFmr$ | $NFmr = (\Sigma\ (GFAr/S_{CP}))/ny$ | Interesting to know the type of buildings affected in the UAD. |
| Mean distance from the CP to the UAD's Historical Centre in a straight line, Dhc (m.) | $Dhc = (\Sigma\ Dhc\ (CPi))/nCP$ | Quantifies the importance of the residential expansion. |
| Mean distance from CP to the coast in a straight line, Dc (m.) | $Dc = (\Sigma\ D_C\ (CPi))/nCP$ | Establishes the proximity of the UAD to the sea. |
| Average slope of the UAD, $SLm$ (degrees) | $SLm$ = mean Slope cells (5 × 5 m) [GIS calculated] | Indicates the type of profile (mountainous, flat, etc.). |
| Built-up area per unit of surface area, $SpGFA$ (m$^2$ of GFA per km$^2$ of UAD) | $SpGFA = GFA / S_{UAD}$ | Rate or intensity of residential construction for the total of the UAD. |
| Risk per unit of surface area, $SpRV$ (€per km$^2$ of UAD) | $SpRV = RV / S_{UAD}$ | Rate or intensity of risk (specific risk) for the total of the UAD. |
| Slope of the straight trend line, $mRR$ (degrees) | See equation (10) | Determines the trend of the RR value in a specific period. |

CP: Cadastral Parcel; S$_{CP}$: surface area of the CP [GIS calculated]; nCP: number of CPs; ny: number of years in the series; S$_{UAD}$ = Urban administrative division Surface area (km$^2$)

**Table 3. Land Susceptibility Index (LSI) values for the classes under consideration**

| Class | Level | LSI interval | No. pixels |
|---|---|---|---|
| 1 | Very low | 10 - 35 | 526 777 |
| 2 | Low | 35 - 60 | 359 376 |
| 3 | Medium | 60 - 95 | 692 863 |
| 4 | High | 95 - 149 | 350 051 |
| 5 | Very high | 149 - 340 | 201 170 |
| TOTAL | | | 2 130 237 |

Source: Cantarino *et al.* (2019)

**Table 4. Probability of occurrence and associated hazard by susceptibility level**

| Class | Level (L) | Level Surface area ($S_L$, has) | Risk Surface area ($S_{RL}$, has) | Spatial Prob. (Ps) | Temporal Prob. (Pa) | Hazard (H) |
|---|---|---|---|---|---|---|
| 1 | Very low | 32 923.6 | 0 | 0 | -- | 0 |
| 2 | Low | 22 461.0 | 0 | 0 | -- | 0 |
| 3 | Medium | 43 303.9 | 25 529 | 0.0005895 | 0.4 | 0.00024 |
| 4 | High | 21 878.2 | 233 675 | 0.0106807 | 0.4 | 0.00427 |
| 5 | Very high | 12 573.1 | 406 913 | 0.0323637 | 0.4 | 0.01295 |
| *Total* | | *133 139.8* | *666 120* | | | |

**Table 5. Vulnerability related to the number of floors.**

| Number of Floors (NF) | Landslide Magnitude (LM) | Building Resistance (BR) | Final Vulnerability (FV) |
|---|---|---|---|
| > 8 | 0.6 | 30% | 0.42 |
| 8 - 4 | 0.6 | 20% | 0.48 |
| 4 - 2 | 0.6 | 10% | 0.54 |
| < 2 | 0.6 | 0% | 0.6 |

**Table 6. Total values and global indicators per municipality. For indicators, means and variation intervals.**

| Name | Value | Name | Value |
|---|---|---|---|
| Total GFA (m$^2$) | 41 642 352 | Total Risk Value *RV* (€) | 5 013 178 |
| Total Risk Ratio *RRt* | 192.5 [0.21 – 869.3] | *Dhc* (metres) | 1 490.1 [95 – 3 954] |
| Mean RR *RRm* | 208.5 [0.3 - 988] | *Dc* (metres) | 8 728.1 [1 047 – 19 466] |
| Mean Risk *RVm* (€) | 2154.7 [31 – 23 252] | *SLm* (degrees) | 16.23 [9.8 – 24.6] |
| *NFm* | 2.09 [1.41 – 5.85] | *SpGFA* (m$^2$ per km$^2$ of UAD) | 26 643 [470 – 170 659] |
| *NFmr* | 1.87 [1.07 – 5.95] | *SpRV* (€per km$^2$ of UAD) | 3388 [ 11 - 37412 ] |
| *mRR Lo* (degrees) | 41.5 [83.7 ~ -89.2] | *mRR Hi* (degrees) | 16.9 [86.5 ~ -87.8] |

**Table 7. Centroids classification values**

| | | | | | Risk Management | |
|---|---|---|---|---|---|---|
| Percentil | Level | Code | SpGFA x1000 | RRt | mRR High | Type |
| 100 - 90% | Very High | A | 170,7 – 131,0 | 821 - 626 | 86 - 80 | Very improvable |
| 90 - 60 % | High | B | 131,7 – 20,8 | 626 - 163 | 80 - 48 | Improvable |
| 60 - 30% | Low | C | 20,8 - 4,5 | 163 - 88 | +48 ~ -40 | Reviewable |
| 30 - 0% | Very Low | D | 4,5 - 0,9 | 88 - 0 | -40 ~ -72 | Suitable |

**Table 8. Clusters centroids and their levels. Organized from A (max) to D (min) according to Table 7**

| | Cluster centroids | | | | | | | Other indicators (mean) | |
|---|---|---|---|---|---|---|---|---|---|
| Cluster number | SpGFA x1000 | Level 1 | RRt | Level 2 | mRR Hi | Level 3 | Cluster CODE | SLm | SpRV x1000 |
| 1 | 170.7 | A | 86 | D | 38 | C | ADC | 11.3 | 14.62 |
| 2 | 152.3 | A | 80 | D | -58 | D | ADD | 13.8 | 12.15 |
| 3 | 69.5 | B | 500 | B | 86 | A | BBA | 14.5 | 34.34 |
| 4 | 62.7 | B | 146 | C | 53 | B | BCB | 10.7 | 9.17 |
| 5 | 81.4 | B | 39 | D | -13 | C | BDC | 10.8 | 3.74 |
| 6 | 20.9 | C | 72 | D | -1 | C | CDC | 6.1 | 0.01 |
| 7 | 20.0 | C | 164 | C | -65 | D | CCD | 14.2 | 1.96 |
| 8 | 18.6 | C | 154 | C | 71 | B | CCB | 14.3 | 2.81 |
| 9 | 0.9 | D | 679 | A | 82 | A | DAA | 22.1(*) | 0.94 |
| 10 | 3.5 | D | 296 | B | 77 | B | DBB | 19.6(*) | 0.93 |
| 11 | 4.4 | D | 88 | D | 54 | B | DDB | 18.8(*) | 0.41 |
| 12 | 1.6 | D | 821 | A | -72 | D | DAD | 23.2(*) | 1.36 |
| 13 | 2.5 | D | 324 | B | -66 | D | DBD | 22.3(*) | 0.79 |
| 14 | 4.5 | D | 105 | C | -39 | D | DCD | 16.7(*) | 0.52 |

(*) Inland hilly areas (SLm >= 17º)

 **Table 9. List of clusters with their characteristics and municipalities assigned. Grouped by intensity construction ratio (SpGFA) from high to low**

| SpGFA | Cluster CODE | Other remarkable Characteristics | Risk building Management | Municipalities |
|---|---|---|---|---|
| Very | ADC | Very Low RR | Reviewable | Benidorm (*) |
| high | ADD | Very Low RR & Trend | Suitable | Calpe (*) |
| High | BBA | *High RR* and Very High growth trend | Very Improvable | Altea, Benitachell |
|  | BCB | High trend | Improvable | Teulada |
|  | BDC | Very Low RR | Reviewable | Alfaz, Xabia (*), La Nucía (*), Denia (*), Villajoyosa, Ondara, Vergel |
| Low | CDC | VLow RR. | Reviewable | Beniarbeig, Benidoleig (*), Benissa (*), Finestrat (*), Gata de Gorgos (*), Orba (*)C |
|  | CCD | VLow growth trend | Suitable | Benidoleig (*), Rafol de Almunia |
|  | CCB | High growth trend | Improvable | Callosa, Polop, Pedreguer, Pego, Sanet y Negrals |
| Very | DAA | *VHigh RR* and growth trend. | Very Improvable | Confrides |
|  | DBB | *High RR* & growth trend. | Improvable | Alcalalí, Benifato, Benigembla, Benimantell, Lliber, Orxeta, Relleu (*), Xaló |
|  | DDB | High growth trend. | Improvable | Bolulla, Castell de Castells, Vall d'Ebo, Murla, Senija, Tormos, Vall de Laguart, Xaló |
| low | DAD | *VHigh RR*. VLow trend | Suitable | Castell de Guadalest, Sella (*) |
|  | DBD | VLow trend | Suitable | Adsubia, Beniardá (*), Tárbena |
|  | DCD | VLow Trend | Suitable | Benimeli (*), Vall de Alcalá (*), Parcent (*), Sagra (*), Vall de Gallinera (*) |

(*) Municipalities with a change in trend from the first part of a series to the second.

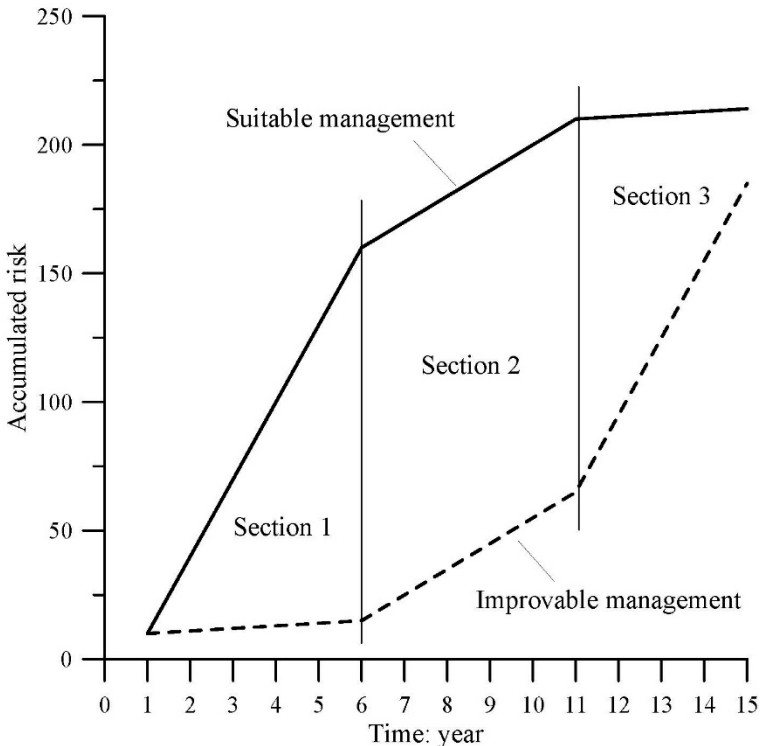

**Figure 1. Theorical evolution of risk accumulated over time for a one-year series pattern**

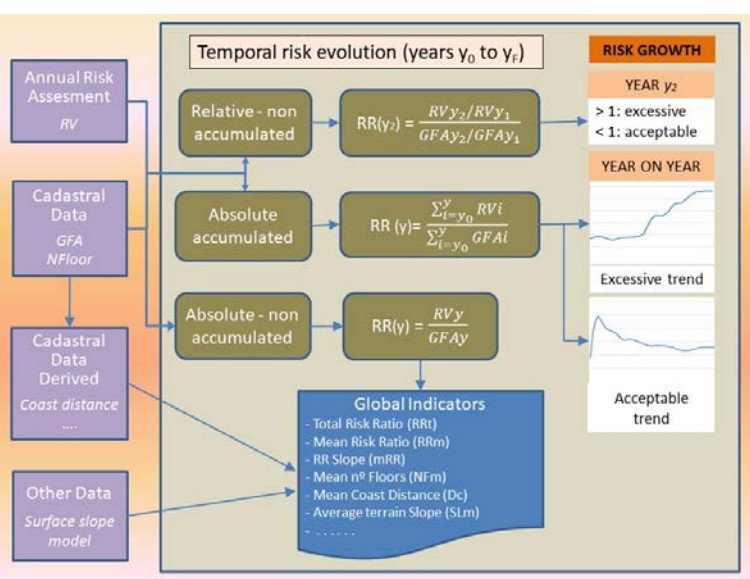

**Fig. 2 Curve trend of different types of Risk Ratio**

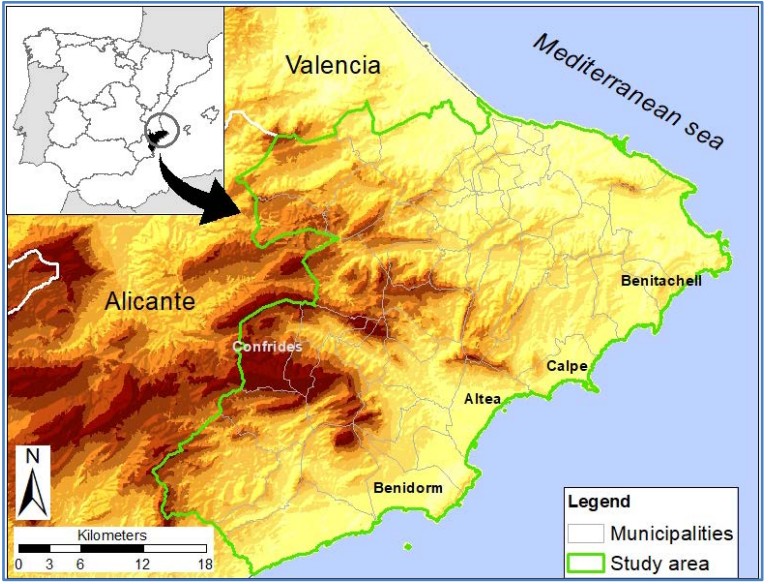


**Fig. 3. La Marina area. Location of some municipalities mentioned in the text.**

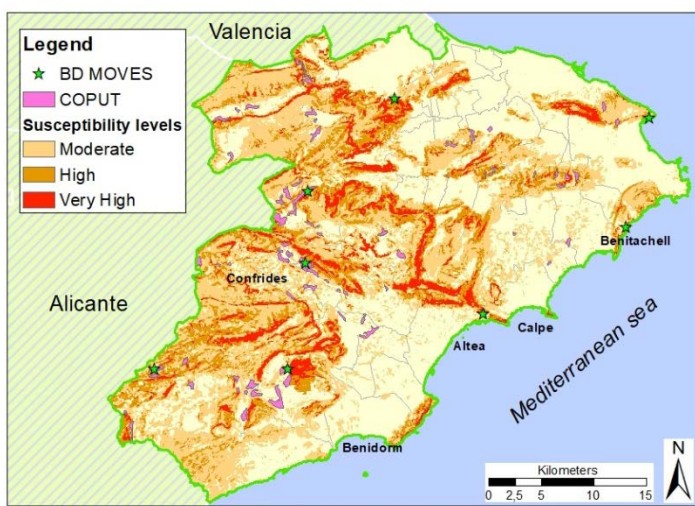

**Fig 4. La Marina area. Susceptibility, landslides location and areas with instabilities**


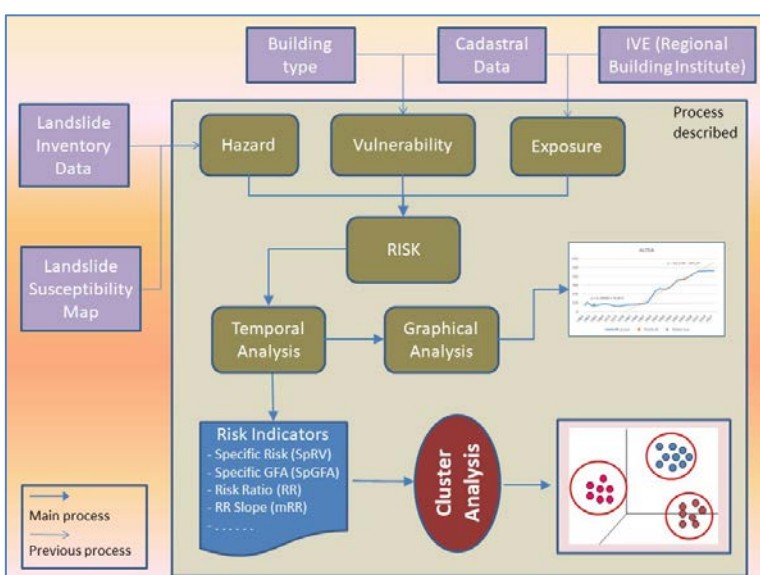

**Fig. 5. Flowchart of the work procedure.**

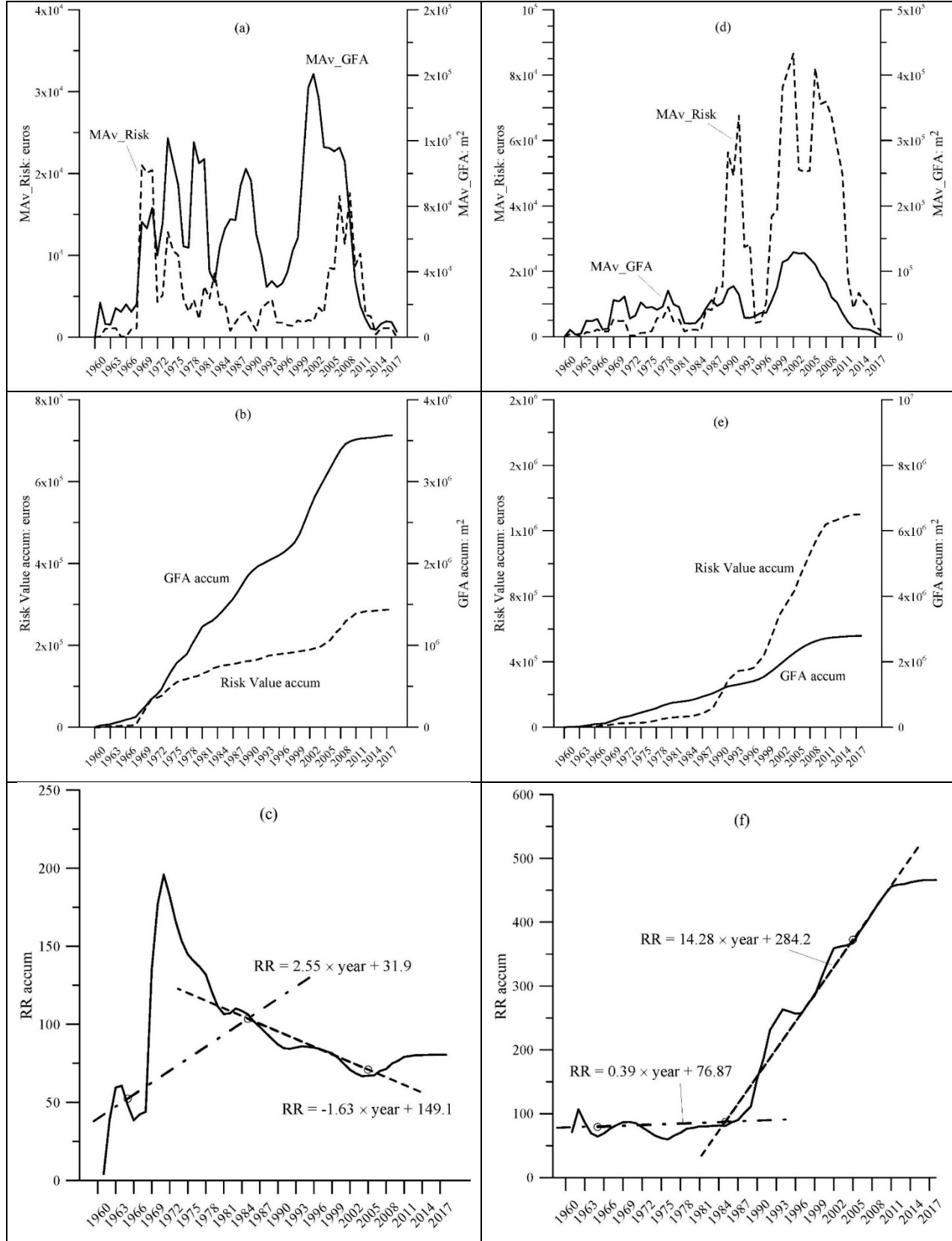

**Figure 6. Evolution of the annual series of GFA, RV and RR in the municipalities of Calpe (a, b, c) and Altea (d, e, f).**


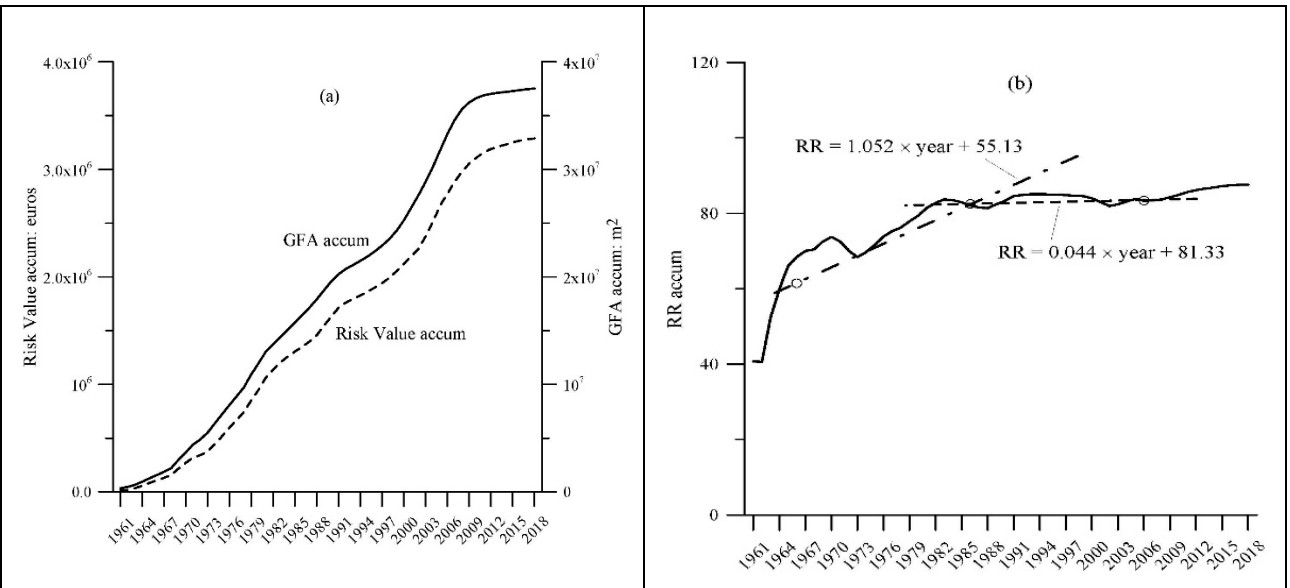

**Figure 7. Evolution of the annual series of GFA, RV and RR for the La Marina area.**

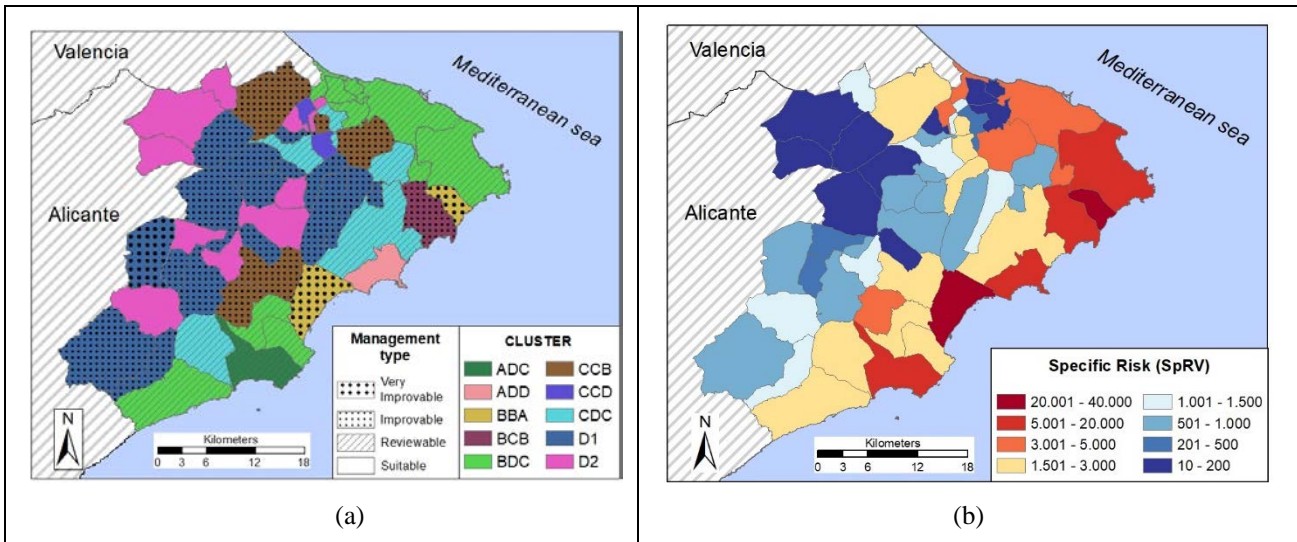

Fig. 8. Map of La Marina: (a) with cluster groups[*], (b) with the SpRV value.

(*) Clusters D1: DAA, DBB, DDB; Clusters D2: DAD, DBD, DCD
