# Peer review of "Landslide risk management analysis on expansive residential areas. Case study of La Marina (Alicante, Spain)"

_Natural Hazards and Earth System Sciences, 2020_

## Referee Comment (RC1) · Paola Salvati (Referee) · 24 Sep 2020

The manuscript concerns the landslide risk issue in relation with the diffusion of residential areas. The aim of the paper is to lay down objective criteria to find how suitable a specific local entity's risk management is by looking at the evolution of its urban development procedures. The authors applied their method in a case study on the Spanish Mediterranean coast as an example of "rural sprawl" generated by second homes and for residential tourism. The final goal of the authors is to determine which is the cause that most affects the increase of landslide risk considering the geomorphological dynamics, the inadequate land management or other random reasons. To evaluate the

landslide risk, they applied the UNDRO general equation of the risk trying to define all the components, partially failing the attempt. To determine the evolution of the residential build-up area they used the annual Gross Floor Area data available for the case study from cadastral parcel data. They then analyzed the temporal distribution of the two elements. The paper addresses the problem of land regulations and possible restrictions in land use according to landslide risk assessment. Local and central administrations can take advantages from the results of their analysis to verify, whether or not, their land regulations are obtaining the right effects. From this perspective, the proposed tool has the right relevance, even if, in my opinion, it is not easily repeatable in other areas given the enormous amount of data (e.g. landslide inventories, landslide temporal series, exposure data, cadastral parcel data, annual data on buildings) needed to apply it, not always available for wide areas. Since the presented tool should be of interest for the scientific community, I would suggest the authors to increase the quality of the paper working both improving the paper structure and in treating some fundamental topics in detail and citing more references.

My individual scientific questions concerns: How do you model the landslide susceptibility? Can you give more details and stress the possible limitations or uncertainties, if they exist, that can affect the results? For the landslide hazard evaluation, you accounted for 8 landslide events. Do you consider the number completely representative of the landslide occurred in the 1,335 km2 of the studied area? Do you account the landslide magnitude considering information on the landslide individual areas? it is not fully clear from the text. Concerning the building physical vulnerability indicators, the cited references refer to debris flows. Is this type of landslide of possible occurrence in the study area? Or do you apply the indicators to other type of landslide? In this case you should explain the reason that support your choice. The cluster analysis section, that in my opinion is relevant for the aim of the paper, should be improved as it is weakly explained and the results poorly described and discussed.

In addition to the comments in the pdf file attached, I add some suggestions to the

paper structure:

Section 2, "General Methodology" needs a graphical schema to illustrate the different theoretical issues. In some cases you refer to results or questions discussed later in the text making difficult the comprehension to the reader. The section has too many sub-chapters and titles interrupting the reading. Section 3, "Case study". It should be better to highlight that you are assessing the specific risk. You are assessing the risk in terms of expected economic loss due to landslide damage to residential building and not the total risk. In my opinion there are some basic data that you missed. It could be useful to add and discuss maps of both the landslide inventory and of the landslide susceptibility model to help the reader to identify the place where the landslides occurred and where the landslide susceptibility give the highest values.

I would suggest a full revision of the English language.

Please also note the supplement to this comment:
https://nhess.copernicus.org/preprints/nhess-2020-216/nhess-2020-216-RC1-supplement.pdf

―――――――――――――――

[Figure]

**Supplement:**

[revised manuscript text omitted]

---

## Referee Comment (RC2) · Eleftheria Poyiadji (Referee) · 14 Dec 2020

The paper discusses the landslide risk and its relation to urban expansion, with or without planning rules related to geohazards. For this reason, a pilot area has been chosen and a quite complicated methodology has been applied, involving landslide inventories, landslide temporal series, susceptibility maps, exposure data, cadastral parcel data, annual data on buildings. The main aim of the study is the identification of objective criteria to find how suitable a specific local entity's risk management is, by looking at the evolution of its urban development procedures. As an idea, "to document that urban expansion without planning (including geological hazards), lead

to an increase of landslide risk" is interesting and if succeed it would be a useful tool in our effort to convince policy makers to take preventive measures by introducing geohazards into urban planning and management. However, the methodology is complicated, as this has been aforementioned, and too many assumptions have been made: For instance, (1) climate change has not been taken into account, a factor that is of the most important for the manifestation of landslides. (2) Although the authors understand that the exposure is different for the various types of landslides it is not clear if they have the data (does the inventory gives details on landslide types) and what type did they finally chose (debris flows?) and why? Moreover, the authors believe that following the proposed methodology would be able to determine what causes the incidence of landslide risk (geomorphology, chance, land management, etc.), and would finally be able to suggest control tools for the public bodies tasked with monitoring such matters. In my opinion the methodology described has many unclear points and many gaps. Finally, I would suggest a major revision of the paper. It is a text not easy to follow, there are many up and downs (see comments on pdf) and points that must be clarified. Also some of the references are not listed at the end of the document.

Please also note the supplement to this comment:
https://nhess.copernicus.org/preprints/nhess-2020-216/nhess-2020-216-RC2-supplement.pdf

**Supplement:**

[revised manuscript text omitted]

---

## Author Comment (AC1) · 3 Feb 2021

**AUTHORS' RESPONSE TO REFEREE No. 1 (Paola Salvati)**

The authors are deeply appreciative of the referee's instructive comments, which have led to our making all the suggested corrections and have greatly helped us to improve the presentation and content of this paper.

Our responses to individual "comments to the author" are given below. Text in green is updated from the original paper's text.

**1. My individual scientific questions concerns: How do you model the landslide susceptibility?**

Landslide susceptibility has been drawn up in keeping with the article mentioned in the text published in *Landslide* (Cantarino et al, 2019). In order to clarify this point, the following paragraph has been added to our article:

"Its characteristics are: pixels of 25 x 25 m as the unit of surface area and the spatial-multicriteria method (SCME) to weight the factors for obtaining the susceptibility values. The three significant factors used were: slope gradient, lithology and land cover."

**2. Can you give more details and stress the possible limitations or uncertainties, if they exist, that can affect the results?**

The main limitation to this study stems from the use of a landslide inventory with few historical records according to the Spanish Geological Survey (BD-Moves) and a territorial analysis carried out in 1999 by the Valencia Regional Government. Both data sources, which are unrelated to this study, should be reviewed and extended by the corresponding official bodies. It is clear that an updated database would provide absolute values different from those obtained in this study. However, we believe that it would not substantially alter the conclusions reached, nor would it invalidate the usefulness of the index proposed in our study.

If necessary, this paragraph can be included in the text.

**3. Do you consider the number completely representative of the landslide occurred in the 1,335 km2 of the studied area?**

It is true that the eight landslides mentioned do not seem to be sufficient to represent such a large area. There have almost certainly been quite a lot more, but they have not been recorded in the inventory by the Spanish Geological Survey. Nevertheless, as mentioned in the previous section, an adjustment to the probability of occurrence in the levels of susceptibility used should not substantially change the trends in the curves obtained.

**4. Do you account the landslide magnitude considering information on the landslide individual areas? it is not fully clear from the text.**

The landslide magnitude (LM) has actually been considered to be constant throughout the zone under study due to the difficulty in calculating it without the necessary field data. This study has followed the criterion of Silva & Pereira (2014) indicated in this paragraph and extracted from their paper: "Taking into account the previous example and the fact that shallow slide characteristics in the study area do not vary too much in terms of affected area, depth of the slip surface, velocity, volume and typical damage, we

assumed a single fixed value for LM. Therefore, the LM was assumed to be 0.6 on a heuristic scale ranging from 0 to 1".

Hence, in our article the following sentence has been modified:
"Therefore, the LM was assumed to be 0.6 for the area of study on a heuristic scale ranging from 0 to 1 (Silva and Pereira, 2014) (see Table 5)".

**5. Is this type of landslide of possible occurrence in the study area? Or do you apply the indicators to other type of landslide? In this case you should explain the reason that support your choice.**

Indeed, the paper cites some authors who have worked with debris flows. However, our zone under study in not especially affected by this type of landslide. The most common hillside instabilities are rockfalls and rotational and translational landslides. This study has only taken into account landslides in keeping with the information available in the inventories according to the Spanish Geological Survey (BD-Moves) and the Valencia Regional Government.

**6. The cluster analysis section, that in my opinion is relevant for the aim of the paper, should be improved as it is weakly explained and the results poorly described and discussed.**

A great effort has been made to improve and clarify this section, as suggested by the reviewer. Table 7 has been reorganised and improved, and a new Table 8 has been drawn up. Both are given below. Finally, the updated text in the article concerning cluster analysis is as follows:

[revised manuscript text omitted]

**7. In addition to the comments in the pdf file attached, I add some suggestions to the paper structure:**

**7.1. Section 2, "General Methodology" needs a graphical schema to illustrate the different theoretical issues.**

Given that the section "Temporal Evolution of Risk" is one of the most relevant ones in the paper, new paragraphs have been added to the article following the reviewer's suggestion. Furthermore, a new diagram has been added (Temporal risk flow chart, Fig. 2) showing the method explained in the section. The updated text is given below, as well as the new figure.

"The adimensional (relative) Risk Ratio (RR) between years $y_1$ and $y_2$ is defined in the following equation:

$$RR(y_2,y_1) = \frac{\frac{RV(y_2)}{RV(y_1)}}{\frac{GFA(y_2)}{GFA(y_1)}} = \frac{rRV}{rGFA} \quad (6)$$

To sum up, it is concluded that $f(y)/g(y)$ is a function whose growth slope is defined by the growth of the Risk Ratio value (RR) for the chosen interval $[y_1, y_2]$. The different options are summed up in Table 1.

It is preferable to use the absolute values from the relationship between RV and GFA in order to be able to compare their magnitudes between the different municipalities. In addition, working with the functions of the accumulated values RVacc and GFAacc, it is ensured that the two base curves are monotonically increasing for the entire period under study. It is easily demonstrated that the quotient function of the accumulated series RVacc/GFAacc also meets the characteristics determined for the RR value in Table 1. These annual values can be transferred to a graph showing the resulting curve in order to analyse its ascending or descending trend (see Fig. 2).

$$RR(y) = \frac{RVacc}{GFAacc} = \frac{\sum_{i=y_0}^{y} RVi}{\sum_{i=y_0}^{y} GFAi} \quad (7)$$

Equation 7 shows the calculation of the accumulated RR values for each year. It is applied for the entire time series available, always starting from an original year $y_0$. In these quotient functions, a simple deterministic trend is going to be assumed."

[Figure]

**7.2. In some cases you refer to results or questions discussed later in the text making difficult the comprehension to the reader.**

To solve this problem, section 2 (with a new sub-section 2.2, "Risk evaluation") and section 3 have been reorganised.

**7.3. The section has too many sub-chapters and titles interrupting the reading.**

An effort has been made to reduce such interruptions in the new text.

**7.4. Section 3, "Case study". It should be better to highlight that you are assessing the specific risk. You are assessing the risk in terms of expected economic loss due to landslide damage to residential building and not the total risk. In my opinion there are some basic data that you missed.**

Good point. Now the text indicates that the type of risk analysed is "economic loss due to landslide damage to residential buildings". Furthermore, an effort has been made to include all of the data necessary to reproduce the calculation made in the paper, although it is possible that some minor data has not been included. However, this should not be relevant enough to alter the results obtained.

Following the reviewer's advice, a new map has been created in Figure 5 attached, indicating the three highest levels of susceptibility, together with the location of landslides according to the Spanish Geological Survey (BD-MOVES) and the areas with instabilities according to the Valencia Regional Government (COPUT).

[Figure]

**7.5. I would suggest a full revision of the English language**

The entire text has been translated by a professional of recognised prestige with a great deal of experience in translating technical, scientific and academic texts. All of the text has been revised again completely and thoroughly. We are willing to improve the text of any paragraph you may indicate in order to achieve the desired quality in writing it. Attached is the certificate of English proof-reading provided by the professional service.

As for the supplementary notes indicated in the article's text, the following modifications have been made:

**Line 20: Please write clearly the first paragraph, give more references and avoid opinions**

That first paragraph has been modified, introducing two new references:

"Landslide risk evaluation, management and mitigation are aspects that have been dealt with profusely in recent decades in the literature specialising in such matters. Specially by the knowledge that a proactive approach to risk management is required to significantly reduce loss of lives and material damage associated with natural hazards (Kalsnes, 2016). There are a multitude of studies on these matters, notably the summary put forward by Dai et al. (2002) with a critical review of landslide research and the strategies for reducing damages and losses, as well as the relevant publications by Lee and Jones (2004) and Glade et al. (2006) with a multidisciplinary perspective on landslide risk management. The recent review of quantitative methods for analysing landslide risk by Corominas et al. (2014) also gives recommendations to improve these procedures.

It is important to consider that the risk associated with landslides is changing as a consequence of environmental change and social developments. Climate change, the increased susceptibility of surface soil to instability, anthropogenic activities, growing (and uncontrolled) urban development and changes in land use with increased vulnerability for the population and infrastructure as a result, all contribute to the change—and in most cases the increase—in the risk of landslides (Gallina et al., 2016)"

**Line 140: Please better emphasize that this is a theoretical example.**

The text has been changed, placing special emphasis on the fact it is referring to a theoretical example, as indicated by the reviewer. The final text is now as follows:

"In the early years of this example situation pattern, […] This theoretical behaviour is shown…"

The title to Figure 1 has also been changed, thus:

"Theorical evolution of the risk accumulated over time for a one-year series pattern"

**Line 333: Please, change "they" with "the"**

The errata has been corrected.

**Line 338. Please, give reference**

The following quote and text have been included in the article:

"These two types of probability—temporal and spatial—are in keeping with equation…"

**Line 343. the information on landslide magnitude??**

The evaluation of the landslide magnitude (LM) has been dealt with in the section on vulnerability.

**Otherwise if the temporal recurrence probability is affected by error, the whole analysis could be affected"**

There is indeed some error in the risk calculation due to uncertainty in calculating the temporal probability. The landslide probability will be based on the historical frequency of recorded landslides, but within reason and making use of this available data, knowledge and experience, too. This leads to a range of estimates, even when using exactly the same basic information. As a result, it was decided not to complicate the calculation method with confidence intervals so as to obtain results that can be interpreted more easily.

**Line 417 "Are debris flows mapped in your study area?"**

No; the only hillside movements mapped in the zone are rotational and translational landslides, as well as rockfalls. This section has been corrected in order to avoid confusion.

**Line 422: please, better explain how you evaluated the BR percentage**

Silva and Pereira (2014) explain the way of calculating the "building resistance" (BR), using a total of five "building features". Recent buildings have very similar construction characteristics (brick walls joined by reinforced concrete) with very little variation in the scope of this study. The only "building feature" with some variation and about which we have data via the cadastre is the number of storeys. Their weighting appears in Table 1 of the aforementioned article by Silva and Pereira. The resistance value has been reduced as much as possible according to this characteristic in order to allow for a greater range of variation in the results for risk calculated.

**Line 433 "It should be remarked as a final reflection on the risk calculation methods, that it does not seem to be essential here to carry out a very comprehensive, exhaustive application".**
**This sentence has no sense. it could be better if you state that it is a simplified method to assess the risk value and find references concerning the application of other simplified methods.**

Our observation refers to the fact the data used is not of such quality as one might wish since there is no official data or specific studies within the scope of this study. Data sources have been used that are not complete or fully up-to-date, which means a lower accuracy on using them in this study. In other words, a complete method has been applied but with a little uncertainty as regards its results, though this does not invalidate the objective or the usefulness of the index proposed in our research. We propose changing the sentence for the following:

"As a final reflection on the application of this or any other method for calculating risk, it should be noted that there is some difficulty in obtaining precise results due to the lack of official data and specific, up-to-date studies in the sphere being studied. Some of these procedures are based on data that is not very exact, and even on subjective evaluations, which means some error must be assumed in the results obtained, though this does not invalidate the objectives or the validity of the index originally proposed in our study."

**Line 460. Please, better explain table 6 in the text**

The following text has been introduced in order to improve the explanation of Table 6:

"The values of these indicators calculated for the 50 municipalities that make up La Marina are shown in Table 6, accompanied by their interval of variation.
A series of annual values were calculated for the 50 municipalities of La Marina area as a whole. The total values for the built-up area (GFA) and risk (RV) are shown in Table 6. The mean values are listed in the same table, as well as their interval of variation of the global indicators in the previous Table 2."

**Line 480. did you first cite and discuss fig.4??**

It is true that Figure 4 is mentioned before naming it. The mention has been deleted since it is not necessary to analyse it in this paragraph.

**Line 524. Why did you not normalized it?**

In the software used to carry out the cluster analysis (Statagraphics), a prior standardisation was carried out on all of the variables used, though it is true that this was not mentioned in the text. It is now mentioned.

**Line 535. Can you show a table with the obtained values?**

As indicated above, the modified Table 7 has now been included as well as a new Table 8 with the results from the cluster analysis (centroids of variables).

**Line 542. Can you please explain what are you referring with segments?**

The term "segment" is perhaps not suitable for a curve; perhaps "curve section" is preferable. The text has thus been changed.

**Line 585. The first three paragraphs 585-600 could be deleted or reduced**

The first three paragraphs of the Conclusions section have been simplified as follows:

"In this vein, it would seem reasonable to think that studies on the mechanics and distribution of landslides, the growth in information about behaviour of the ground, the restrictions imposed on residential expansion etc. should progressively improve the effectiveness in tackling the risks. However, it has been shown that not all municipalities are capable of reducing the incidence of these risks over time and that, according to Fig. 5, this incidence is still generally high. So why is this happening?"

**Certificate of translation of the paper "Landslide risk management analysis on expansive residential areas. Case study of La Marina (Alicante, Spain)" (ES>EN)**

Translator: Gary Smith

Tax ID: (ES) X1600081V (Professional activity group 774 of the Spanish Tax Agency)

I, Gary Smith, independent translator with tax no. (ES) X1600081V, voting board member of the International Association of Professional Translators and Interpreters (IAPTI) and member of ASETRAD, the Mediterranean Editors and Translators Association (MET) and La Xarxa (XarxaTIV, former president) certify that I have translated and revised the text "Landslide risk management analysis on expansive residential areas. Case study of La Marina (Alicante, Spain)" from Iberian Spanish into British English and I testify to the standard of English used therein.

Gary Smith

3 February 2021

[Figure]

**De conformidad con lo que establece la Ley Orgánica 15/1999 de Protección de Datos de Carácter Personal, y con su Reglamento de Desarollo, y con el Reglamento General 2016/679 del Parlamento Europeo y del Consejo, de 27 de abril de 2016 de Protección de Datos, el RGPD (UE), le informo que los datos personales arriba serán incorporados a un fichero bajo la responsabilidad de GARY SMITH, con la finalidad de poder atender los compromisos derivados de la relación profesional que mantengo con usted y con Hacienda. Puede ejercer sus derechos de acceso, cancelación, rectificación, oposición, portabilidad, información y restricción del tratamiento mediante un escrito a la dirección: Calle Lanzarote 3, 5, CP 46011 Valencia. Si en el periodo de 30 días no me comunica lo contrario, entenderé que sus datos no se han modificado, que se compromete a notificarme de cualquier variación y que tengo su consentimiento para tratarlos a fin de poder tramitar su facturación y mantener nuestra relación profesional.**

---

## Author Comment (AC2) · 3 Feb 2021

**AUTHORS' RESPONSE TO REFEREE No. 2 (E. Poyiadji)**

The authors are deeply appreciative of the referee's instructive comments, which have led to our making all the suggested corrections and have greatly helped us to improve the presentation and content of this paper.

Our responses to individual "comments to the author" are given below. Text in green is updated from the original paper's text.

**1. The methodology is complicated, as this has been aforementioned, and too many assumptions have been made. For instance, (1) climate change has not been taken into account, a factor that is of the most important for the manifestation of landslides.**

Indeed, the effect of climate change has not been specifically taken into account in our study. We are aware that climate change increases the susceptibility of surface soil to instability due to agricultural areas being abandoned, forest fires and an increase in heavy precipitation. However, it is difficult to quantify its impact in a deterministic study and to thereby discard its effect. We believe it is acceptable to assume that climate change does not appreciably affect the susceptibility values obtained in this study, as indicated in the paper's text (see line 105). Nevertheless, its effects on other data sets used in the paper are implicitly included and reflected in the trend curves obtained.

Even so, in order to not overlook the significant role played by climate change in hillside instability, the following paragraph has been included in the paper:

"It is important to consider that the risk associated with landslides is changing as a consequence of environmental change and social developments. Climate change, the increased susceptibility of surface soil to instability, anthropogenic activities, growing (and uncontrolled) urban development and changes in land use with increased vulnerability for the population and infrastructure as a result, all contribute to the change—and in most cases the increase—in the risk of landslide (Gallina et al., 2016)"

It is also a pity that there are so few studies on landslides in our country that look in detail at the effects of climate change, as pointed out by Gariano and Guzzetti (2016, cited in the paper): "Spain do not consider landslides in their climate change adaptation strategies, or in related preparatory and accompanying reports".

**2. Although the authors understand that the exposure is different for the various types of landslides it is not clear if they have the data (does the inventory gives details on landslide types) and what type did they finally chose (debris flows?) and why?**

Unfortunately, the inventory used is not very precise and only differentiates rockfall from other kinds of landslide. This data source comes from a study carried out by the Valencia Regional Government (COPUT), in which we did not take part. In our zone, landslides are generally rotational or translational, in addition to rockfalls. Debris flows are not very common and are not specifically differentiated in the aforementioned inventory. That is why this study mainly concentrates on landslides without considering rockfalls or debris flow.

**3. Moreover, the authors believe that following the proposed methodology would be able to determine what causes the incidence of landslide risk (geomorphology, chance, land management, etc.), and would finally be able to suggest control tools for the public bodies tasked with monitoring such matters. In my opinion the methodology described has many unclear points and many gaps.**

A great effort has been made to improve and clarify this section, as suggested by the reviewer. Table 7 has been reorganised and improved, and a new Table 8 has been drawn up. Both are given below. Finally, the updated text in the article concerning cluster analysis is as follows:

[revised manuscript text omitted]

$$RR\ (y) = \frac{RVacc}{GFAacc} = \frac{\Sigma_{i=y_0}^{y}\ RVi}{\Sigma_{i=y_0}^{y}\ GFAi} \qquad (7)$$

Equation 7 shows the calculation of the accumulated RR values for each year. It is applied for the entire time series available, always starting from an original year $y_0$. In these quotient functions, a simple deterministic trend is going to be assumed."

[Figure]

**4. Finally, I would suggest a major revision of the paper. It is a text not easy to follow, there are many up and downs (see comments on pdf) and points that must be clarified.**

Following the reviewer's advice, sections 2 and 3.3 have been completely restructured.

**5. Also some of the references are not listed at the end of the document.**

The mistakes in the bibliographical citations have been corrected.

As for the supplementary notes indicated in the article's text, the following modifications have been made:

**Line 52. Missing reference Di Martire 2012**

This reference has been included.

**Line 105: Too many assumptions**

This paragraph's text has been changed in order to explain the reasons for not specifically including the effects of climate change in this study. The modified text is:

"The second data set must arise from the geolocalised map of risk distribution. Normally, this is based on a landslide susceptibility map (LSM) that has been deemed stable during the period analysed. Indeed, the risk map is calculated based on the temporal nature of construction and must be approximately in sync with this process. Moreover, the occurrence of a landslide is generally linked to trigger mechanisms that respond to events subject to a specific return period. The probability calculation also uses feedback from the appearance of these events, whose frequency is being modified as a result of climate change. However, according to Gariano & Guzzetti (2016), the effects of climate change on the type, extent, magnitude and direction of the changes in the slopes' stability conditions, and on the location, abundance and frequency of the landslides, are not completely clear. In the end, climate change is not going to be taken into account specifically in this work."

**Line 134:**

This sentence has been changed for the following one:

"The essential purpose of this work is to define a reliable, simple method that will enable the risk's dynamics to be described."

**Line 137: "Th e", errata**

Corrected

**Line 156: too much up and downs trying to look for the right equation**

True. As indicated above, the text has been reorganised in sections 2 and 3.3 to smooth out these "ups and downs".

**Line 244:    Total risk value?? Due to landslides or any other geohazard .Please clarify**

This sentence has been changed for the following text:

"The work by Cantarino et al. (2014) emphasises that Alicante was the province most affected by landslide risk value on residential buildings in the Valencia Community Region (Spain)".

**Line 256:    Needs to be modified. Unclear sentence: "Its mountainous terrain means the coastal strip is not exempt from risk, a situation that is aggravated by its high value for tourism and residential occupation".**

This sentence has been changed for the following text:

"Its extensive mountainous orography reaches the coastal strip itself, which is not free from risk. This situation is aggravated by being highly attractive for tourism and its residential occupation."

**Line 260: Missing reference EFA, 2006:**

EFA corrected for EEA

**Line 272: Instituto Geológico y Minero de España: in English : Spanish Institute of Geology and Mining or Spanish Geological Survey**

The text has been changed for: "Spanish Institute of Geology and Mining"

**Line 298: "In this study, the aforementioned thresholds are used to evaluate the risk in different cadastral parcels at any moment, as well as to determine their evolution over time and finally to calculate the level of hazard".**
**Unclear paragraph. How susceptibility will lead to risk and then in hazard??**

This paragraph has been changed for the following text:

"For this study, the spatial probability for each class has been determined by comparing these susceptible areas with the ones indicated in the inventory. This information, together with the temporal probability, has enabled the hazard and finally the risk to be calculated."

**Line 333: They year of…, errata**

Changed to *"The year of …."*

**Line 498: "The possible explanation for this has been given above."**
**It is not an easy text. There are repeated up an downs**

In keeping the reviewer's suggestion, the following text has replaced the previous one, removing the need to look for this explanation in a section already dealt with.

"The possible explanation could be that the plots at greatest risk of landslide begin to be used at a greater pace once the best plots have been occupied following a period of intensive building activity. In other words, it is possible that when suitable plots become scarce, the next buildings are constructed in a worse location and thus a greater risk is taken on."

---

## Author Response (AR2)

**NHESS April 2021, 2nd review**

**REFEREE #1 RESPONSE**

**In this review of the article, we have taken into account all of the recommendations given and made the following changes:**

The paragraph 3.2 "Implementation of the method" describes how the method was applied to the case study, and contains the workflow of figure 5 which, upon careful reading, shows as part of the presented procedure a cluster analysis, which is explained in the following paragraph of discussion. Indeed, much of the discussion section lists the results of the cluster analysis leading to two negative consequences:

1) the cluster analysis is weakly explained, important details are not provided, even if listed in Tables 7 and 8, they are not sufficiently described in the text,

**We have included more extensive explanations that are more pertinent to the cluster analysis carried out, giving reasons for the number of groups used. We also give details of the procedure for regrouping into four different levels for the values of the centroids calculated by the cluster analysis. All of this is summed up in the new Table 7 entitled "Centroid classification values".**

2) the discussion paragraph suffers from lack of arguments. In particular, I can't understand how the cluster codes listed in table 7 are transformed into three classes of risk building management (improvable, reviewable, suitable). I suggest a revision of the two paragraphs.

**Indeed, the procedure for assigning the types of management shown is better displayed now in the new Table 7. The definition for these types of management is based on the slope in degrees of the straight line calculated for the upper (or most recent) section in the graph of annual evolution in RR (mRRHi variable); in other words, the one corresponding to years before the 1980s. It was not well executed in the article's original table so we have modified it, correcting some values.**

**We recommend reading Table 7 again to better understand the solutions provided to the suggestions proposed.**

**The other two errors noted in the text have also been corrected.**

**REFEREE #3 RESPONSE**

**In this review of the article, we have taken into account all of the recommendations given and made the following changes accordingly:**

1) The introduction section underline the role played by a suitable risk management and in particular of a suitable risk building management. However, this part of the story is a bit neglected in the interpretation of the results. I strongly suggest to specify better the focus of the analysis since the introduction and to enforce the results' discussion part accordingly.

**The following paragraph has been introduced at the end of section 1:**

*Another significant aim of this work should also be noted: This involves differentiating correct management of the terrain (specifically addressing its occupation by residential housing) from management that can clearly be improved. In particular, considering the risk of landslide for residential housing, the possibility of said risk becoming stabilised is studied over the time series. In this case, the management can be deemed adequate.*

*Nevertheless, if the risk increases over time, then it can be attributed to improper management, which should be corrected. The aims of this work also include analysing this situation, as well as determining what causes an increase in landslide risk, for example by considering geomorphological dynamics, inadequate land management, even bad luck, etc.*

2) On the same line of point 1, it would be interesting to underline that the story described on Figure 1 is only one of the potential scenarios. In fact, would be interesting to present another figure where another urban sprawl model is presented (e.g. it is possible to argue that more safer places are occupied at the very beginning of the urban growth process while during the urban sprawl less convenient places in terms of landslide risk, or general risk, are used). This would give the chance to introduce the pivotal role of landslide risk management in the growth of these territories.

**Indeed, the pattern of growth in risk considered to be non-suitable or improvable would be a good contribution. This new theoretical growth pattern has thus been included in Figure 1, accompanied by the following text below the figure.**

[Figure]

**Figure 1. Theoretical evolution of risk accumulated over time for a one-year series pattern**

*However, a varying panorama of unsuitable or improvable risk can also be found (Fig. 1). This type of growth in risk can arise when the pressure to build residential housing is so great that spaces become occupied that do not have the optimal conditions in terms of location and which until then had maintained their natural characteristics. Building on such spaces may entail taking greater risks because safer terrains have already been used up. Hence, the great increase in risk in Section 3 (Fig. 1, "Improvable management" line), should not be admissible in proper territorial management, and it is thus essential to provide tools to demonstrate such anomalies as shown in this work.*

3) I have some concerns on the cluster analysis. In particular, there is a lack of details. As an example, why the choice of 14 clusters? Finally, the organization in the four groups A, B, C, D is not clear too. On the same line, please provide more details on the sample selection and in the application of the cluster analysis.

**It is true that establishing the number of clusters is not simple, nor are there standard procedures to solve this, but it is often solved by simply applying common sense. In this case, we have used a minimum of 10 clusters as the basis in order to adequately include the high and low interval of variation of the three variables considered in the analysis. It should also be mentioned that we are certain there are two cases that must belong to different clusters. These are the two municipalities with the greatest rate of construction, which clearly have different behaviours in their annual variation in risk: Benidorm with mRRHi= +38°, and Calpe with mRR Hi = -38°. Based on 10 clusters, trial and error calculations have been carried out upwards. The two are separated when 14 clusters are reached and we understand that this number is suitable because it is accompanied by a definition that is congruent with the rest of the groups.**

The clusters have been organised by grouping them by greater or lesser value of their centroids into a series of levels. Four levels have been established: A (very high), B (high), C (low) and D (very low), which are obtained by applying the percentiles of 90, 60, 30 and 0% of the series of one-dimensional centroids. These limits thus established are particularly intended to restrict the upper values of the series (percentile > 60% in A and B). It is thus possible to more clearly highlight the cases that should be addressed in order to manage risks properly. Four types of risk management evaluation have been defined, taking into account the mRRHigh value (final section of the slope of the straight trend line).

The following table specifies these intervals and is now included in the text as Table 7 ("Centroid classification values"):

| | | | | | Risk Management | |
|---|---|---|---|---|---|---|
| Percentile | Level | Code | SpGFA × 1000 | RRt | mRR High | Type |
| 100 - 90% | Very High | A | 170.7 – 131.0 | 821 - 626 | 86 - 80 | V. improvable |
| 90 - 60 % | High | B | 131.7 – 20.8 | 626 - 163 | 80 - 48 | Improvable |
| 60 - 30% | Low | C | 20.8 - 4.5 | 163 - 88 | +48 ~ -40 | Reviewable |
| 30 - 0% | Very Low | D | 4.5 - 0.9 | 88 - 0 | -40 ~ -72 | Suitable |

Lastly, greater explanations and details have been added to the text regarding the results of the cluster analysis and how they were obtained.

4) The other major point is related to the description of the results. In the present version, this part is hard to digest. In order to improve the readability of the section it would be interesting to present results in Figure 6 aggregated also for municipalities with suitable risk building management with respect to the non-suitable municipalities

We agree that the interpretation of the results has been particularly complicated on dealing with real cases calculated for 50 municipalities with very different growth patterns in RR. Figure 6 shows two municipalities with a high construction rate near the coast but with very different behaviour as regards the growth of risk in residential buildings (Altea with mRRHi= +85° and Calpe with mRRHi = -38°). We understand that these are significant extreme examples and that they should be shown apart due to their unique nature. We consider that completing these graphs with aggregated examples will not provide greater information and may hinder their individual interpretation, so we have preferred to maintain the original arrangement.

However, in the results section we have introduced new references to Fig. 1 in order to better explain the curves shown in the aforementioned Fig. 6. These references are intended to improve comprehension of this section, since they deal with general patterns recognisable in the municipalities mentioned.

Explanations in the section devoted to discussion of the results have also been modified and simplified to improve comprehension, deleting complex references and making them more readable. The text has been adapted to the changes made in the previous section. Fig. 8a has been suitably adjusted.